# Training University Psychology Students to Teach Multiple Skills to Children with Autism Spectrum Disorder

**DOI:** 10.3390/bs15060742

**Published:** 2025-05-27

**Authors:** Daniel Carvalho de Matos, Ryan Matos e Silva Moura de Brito, Fabrício Brito Silva, Juliana Ribeiro Rabelo Costa, Leila Bagaiolo, Claudia Romano Pacífico, Pollianna Galvão Soares

**Affiliations:** 1Postgraduate Program in Psychology, Campus do Bacanga, Federal University of Maranhão, São Luís 65080-805, MA, Brazil; rabelo.juliana@discente.ufma.br (J.R.R.C.); galvao.pollianna@ufma.br (P.G.S.); 2Center for Teaching, Research in Psychology, Inclusive Education and Health, Evoluir Institute, São Luís 65075-690, MA, Brazil; 3Laboratory for Assessment, Research and Intervention in Autism Spectrum Disorder, Campus Renascença, CEUMA University, São Luís 65075-120, MA, Brazil; ryan028640@ceuma.com.br (R.M.e.S.M.d.B.); fabricio.brito@ceuma.br (F.B.S.); 4Gradual—Behavioral Intervention Group, Sao Paulo 05458-000, SP, Brazil; leila.bagaiolo@grupogradual.com.br (L.B.); claudia.romano@grupogradual.com.br (C.R.P.); 5Lato Sensu Postgraduate Course “Applied Behavior Analysis to Autism and Other Neurodivergences”, Pontifical Catholic University of Sao Paulo, Monte Alegre Campus, Rua Monte Alegre, 984, Perdizes, Sao Paulo 05014-901, SP, Brazil

**Keywords:** behavioral skills training, university participants, children with ASD, discrete trial teaching, skills

## Abstract

Training people interested in implementing Applied Behavior Analysis (ABA) interventions to children with autism spectrum disorder (ASD) is important to promote skill gains. A recommended training package is called behavioral skills training (BST), which involves four components (didactic instruction, modeling, role-play, and performance feedback). Background/Objectives: The purpose was to assess the effects of BST on the accurate teaching of multiple skills via DTT by six psychology university students to a confederate and six children diagnosed with ASD. Generalization and maintenance assessments were conducted. Results: Through the research conditions, all university participants were able to teach ten different skills (sitting still, motor imitation, making requests, vocal imitation, receptive identification of non-verbal stimuli, making eye contact, following instructions, intraverbal, labeling, receptive identification of non-verbal stimuli by function, feature and class) with a high integrity level to the children. In addition, across four months after training, all participants maintained high teaching integrity levels while teaching skills to the children related to their individualized curriculum goals. Each child accumulated over 1000 correct responses across several sessions. The university participants rated their training with the highest possible score in a social validity assessment. Conclusions: BST successfully trained psychology university students to accurately teach multiple skills via DTT to children with ASD and involved long lasting effects. Limitations and new avenues for research were discussed.

## 1. Introduction

Principles and procedures from Applied Behavior Analysis (ABA) are frequently used to teach socially relevant behavior change ([4]; [13]; [24]). ABA represents one of the three branches of the science of behavior analysis (Behaviorism; Experimental Analysis of Behavior; Applied Behavior Analysis). ABA focuses on changing socially significant behavior ([25]). Leaf et al., for example, recently organized a handbook of ABA-based interventions for autism spectrum disorder (ASD). It provided knowledge for professionals, parents, and other caregivers and individuals diagnosed with ASD on evidence-based practices (EBPs) in the field of behavior analysis for ASD. Across the chapters, among several EBPs, two were of major interest for the current study: discrete trial teaching (DTT) and behavioral skills training (BST). Scientific evidence of EBP is systematized in literature reviews and informed through international reports. The National Professional Development Center on Autism Spectrum Disorder (NPDC) and the National Clearinghouse on Autism Evidence and Practice (NCAEP) investigate and disseminate EBP practices (https://ncaep.fpg.unc.edu/; https://autismpdc.fpg.unc.edu/. Accessed on 15 May 2025). The latest report, published in 2020, revealed a total of 28 EBPs, which were committed to boosting the development of individuals diagnosed with ASD through the modification of socially significant behaviors, that is, by decreasing undesired interfering behaviors and establishing more adaptive ones, including appropriate language and communication ([36]; [40]). ASD frequently involves deficits in language and communication and the presence of stereotypic/repetitive behavior patterns ([1]). DTT is recommended for learners who need comprehensive interventions to target multiple-skill acquisition ([24]; [28]; [39]; [42]).

DTT is usually applied in a structured environment with control of possible distracting stimuli. When a trial to teach a skill is about to be administered, the interventionist first needs to evoke the learner’s attention and make sure that all necessary materials are available. When the instruction of the trial is provided, some seconds are allowed for the emission of a response by the learner. If the response is correct, a reinforcer, previously defined through a stimulus preference assessment procedure, is provided. If an error occurs (or no response is emitted during the allowed time), a correction procedure is applied (the correct response, for example, is revealed to the learner). Prompts may also be used along the process to increase the likelihood of independent performance. The termination of the trial is followed by a brief interval until a new teaching trial is presented. During the provision of DTT, the learner is exposed to different targets as he or she meets arbitrary learning criteria ([24]; [39]; [42]).

The literature has discussed the importance of practitioners incorporating compassionate and humane care into their ABA practices to learners with ASD, including DTT. It has been argued by the [7] ([7]) that behavior analysts should involve the families in the intervention plans with compassionate care and empathic responding. Collaborative relationships may strengthen adherence to interventions, empowering the families. One way through which behavior analysts may achieve this outcome is by avoiding the use of technical jargon in the field, since terminology commonly does not correspond with the typical vernacular used by people in general who are not behavior analysts. Compassionate care includes providing support and explanations in adapted language that is easy for the more general public to understand. Recently, [32] ([32]) developed a tool for assessing compassionate collaboration with families. They discussed that the tool may help identify skills that behavior analysts should acquire to improve intervention outcomes, intervention adherence, and perception of service providers. The tool consisted of a three-point scale questionnaire to check whether the interventionist had 25 skills on compassionate care. Questions referred to the domains of collaborative approaches and use of proper language and communication with families/clients. The authors hoped that their tool would help practitioners within the field of ABA to assess and improve their provision of compassionate interventions.

In Brazil, the number of well-trained professionals is still not sufficient for the growing number of children with ASD who need intensive behavioral interventions ([3]; [31]). An alternative to this could be training parents and other caregivers to implement interventions to their children themselves, which is also EBP ([36]). [23] ([23]) states that parent training may promote improvements in the relationship with their children with ASD. It also increases the intensity of interventions, which is important for learners who require a comprehensive approach. Parent training also influences the establishment of generalization and maintenance of targets over time. It helps to improve the parents’ optimism as for the future of their children. It induces positive feelings about development and reduces intervention costs, requiring less participation from professionals. However, involving parents and other caregivers in the process of implementing interventions through DTT (e.g., [3]; [17]; [22]) and other more naturalistic formats (e.g., [5]; [14], [15]; [37]; [41]) may not be an easy task in Brazil. Many families have a low socioeconomic status, the members spend many hours at work or on household chores and are often unavailable to be trained ([3]; [20]). Furthermore, family members, especially parents of children with ASD, commonly find themselves under high levels of stress in these circumstances, which can make it even more difficult for them to adhere to a training process. Training programs for a greater number of professionals are warranted to expand the possibilities of access to quality and lower-cost behavioral interventions for children with ASD, whose family members lack the time to be properly trained.

A recommended training package for people interested in carrying out behavioral interventions using a format such as DTT is called BST. It consists of four components: didactic instructions to implement the interventions properly; modeling (two actors, representing interventionist and child with ASD, rehearse the implementation of DTT trials, or the trainer directly rehearses with a child); role-play (the person being trained implements DTT to a confederate/actor or child with ASD); performance feedback by the trainer concerning the correct or incorrect implementation of DTT. Research reveals that BST has been used in training professionals, university students, and caregivers (e.g., parents) to implement DTT in learners with atypical development. Research has demonstrated the acquisition of repertoires in different domains, highlighting, for example, pairing identical non-verbal stimuli such as objects and pictures; selecting a variety of cards with printed words, images of common stimuli and objects under the control of their corresponding dictated names; oral reading of printed words and labeling objects and pictures; motor imitation; following instructions to perform simple actions; vocal imitation; answering questions and other verbal stimuli; making verbal requests; sitting still and making eye contact; the emission of sentences to describe non-verbal stimuli; the dictation and copying of words; addition and subtraction; the narrative retelling of stories and answering comprehension questions ([3]; [9]; [16]; [21]; [22]; [27]; [30], [31]; [33], [34]; [38]).

In some of the investigations, during training, the participants always interacted directly with children with ASD and other cases of atypical development to teach skills ([9]; [27]; [33], [34]). In other studies, the participants interacted during training with a confederate who pretended to be a child with ASD ([16]; [21]; [22]; [30], [31]; [38]). Overall, in all studies shown, BST was successful in the sense that the participants, either professionals or university students, learned to teach several skills accurately. There were cases with evidence of the generalization of accurate teaching of skills to children with ASD, based on training DTT rehearsal with a confederate ([22]; [21]; [30]; [38]). In all studies, increases in performance accuracy (or teaching integrity levels) referred to the correct implementation of several DTT components (e.g., the interventionist evoked the learner’s attention, provided instruction consistently, specified the expected response and without repetition, only reinforced the learner’s correct response, etc.) by each participant across the presentation of trials to teach skills. The more the components were implemented correctly, the more accurate the teaching of skills was.

Although previous research into BST demonstrated its effectiveness in training individuals to conduct DTT accurately for children with ASD as learners, there are still limitations that justify carrying out new investigations. A recent meta-analysis conducted by [17] ([17]) resulted in the identification of 46 studies, involving single-case research designs, which assessed the effects of BST, or any of its four components (didactic instruction, modeling, role-play, and feedback), on the implementation of DTT with accuracy/integrity by different participants. In the studies, the participants were either parents, professionals, or paraprofessionals. It was found that when the BST components were implemented together in training, they were statistically significantly effective and that, on average, the participants from the studies in the meta-analysis demonstrated teaching integrity with 96.06% of DTT components implemented correctly. On the other hand, it is important to mention that the studies from the meta-analysis did not include consistent information on data from the learners with ASD. The focus was more specifically on teaching integrity data from those people who were trained through BST to implement DTT. More research on BST is warranted to assess its effects on systematic skill acquisition by learners with ASD through DTT. The current research sought to do this.

Moreover, it is also important to emphasize that previous studies on BST, involving single-case designs, assessed effects on the accurate teaching of few skills. [33] ([33]), for example, trained their participants to solely teach the pairing of identical non-verbal stimuli, whereas [34] ([34]) trained to teach the same skills but also assessed the generalization of accurate teaching of selecting cards with printed words under control of their dictated names. More recently, [22] ([22]) solely trained their participants to accurately teach the labeling of pictures. [3] ([3]) trained their participants to teach four different skills, including cases such as sitting still, describing pictures through sentences, answering questions and reading syllables. According to [20] ([20]), training people to teach multiple skills may produce more significant gains for learners with ASD. They conducted a longitudinal study and did not use a single-case design. In this previous research, caregivers, who were trained to systematically teach at least ten different skills through DTT across a year, produced significant developmental gains for their children with ASD, confirmed by standardized development assessment protocols. However, caregivers who were trained and supervised to teach fewer skills produced less impactful developmental gains for their children. This suggests that training people to teach multiple skills through DTT may predict more significant gains and this should also be investigated using single-case research methodology. Thus, one of the research questions of the new investigation consisted of the following: (1) Did BST effectively train university students to accurately teach multiple skills via DTT to a confederate pretending to be a child with ASD?

In addition, considering that the assessment of generalization is a predictor of the effectiveness of an intervention, once BST proved successful, the second research question consisted of the following: (2) Did BST produce the generalization of accurate teaching of multiple skills, including new ones not covered in training, to children with ASD across post-training probes? Finally, considering that the previous studies did not clearly assess the impact of BST on performance accuracy by the university students and the acquisition of skills by the children in the long term, that is, across several months, the final research question consisted of the following: (3) After BST and assessment of generalization proved successful, would the high teaching integrity levels by the university students, during the provision of DTT and acquisition of skills by the children, be demonstrated across four months on average? It was hypothesized that all the guiding questions of the research would be confirmed.

## 2. Materials and Methods

### 2.1. Participants

Six undergraduate psychology students (P1–P6) participated in the research, with ages ranging between 20 and 23 years old. None of them showed previous experience with ABA-based interventions, including DTT, to children with ASD, which represented an inclusion criterion. Participants with previous knowledge were excluded from the study. The selected students were defined as interns at the university laboratory where data collection was carried out. They were referred to by the coordinator of the university’s psychology course. They voluntarily participated in this research. All of them showed interest in the activities developed in the laboratory, but none of them had access to services provided to children with ASD until their training through research had been completed. Six children with ASD (CD1–CD6), aged between 5 and 9 years, also participated as learners in DTT-based interventions. They all received interventions once a week for 1 h and 30 min in the university-based laboratory regardless of the current research. Each child’s individualized ABA-based curriculum is not presented in this method section, but since the psychology students specifically taught skills from each child’s curriculum during the last condition of the study (maintenance), the skills are presented in the Results section. To be eligible to participate, the children could not demonstrate severe behaviors related to hetero-aggression, self-injury, and property destruction.

### 2.2. Environment

Data collection was carried out in a university laboratory room. It contained a table and two chairs. At different stages of the research, a confederate, who simulated behaviors of a child with ASD, sat on one of the chairs. Sometimes, either a psychology student or an experimenter sat on the other chair, facing the confederate, to provide DTT trials. The experimenter conducted trials when he needed to demonstrate the correct teaching of skills. Each participating student conducted DTT trials under assessment and BST conditions to establish a more accurate teaching of skills to the confederate. In other situations, designed to measure the generalization and maintenance of the accurate teaching of skills over four months, each of the six children with ASD sat on the chair that previously accommodated the confederate.

### 2.3. Data Collection Instruments

Firstly, the definition of the skills taught by the university participants was based on possible curricular goals commonly established to many learners diagnosed with ASD in a manual of ABA-based interventions ([29]). When the research began, the participants used recording sheets to measure skill gains attained by a confederate and children with ASD involved. The research team also used recording sheets to measure the correct and incorrect implementation among 11 possible DTT components, which could be demonstrated across the administration of each of ten trials per session to teach a given skill. Table 1 shows each of the 11 DTT components.

Part of the stimuli used to teach skills by the university participants was organized in plastic cards, measuring 6 × 3 cm. The cards comprised images representing several categories such as animals and transportation. Whenever a learner emitted a correct response across DTT trials, praised and access to a preferred game or toy were provided.

### 2.4. Dependent Variable and Independent Variable

In this investigation, the main dependent variable (DV) consisted of the number of DTT components implemented correctly by the university participants during the teaching of skills to the confederate and child with ASD. As mentioned before, each participant could implement among 11 possible components during the provision of each teaching trial in a session. The independent variable (IV) consisted of the implementation of four BST components (didactic instruction, modeling, role-play, and performance feedback) by an experimenter. During BST role-play, across trials with the emission of correct responses by the confederate learner, the implementation of a correction component by the university participant was not possible.

Across trials with the emission of incorrect responses by the confederate, the implementation of a reinforcement component by the university participant was not possible. In a trial to teach the skill to follow an instruction, for example, when the participant said, “clap your hands” and the confederate raised the hands, the participant had to implement a correction procedure component, that is, C8. The C7 component of providing a reinforcer would not be possible. The data were organized in a percentage format per session, that is, the number of DTT components implemented correctly was divided by the total number of possible components, and the result was multiplied by 100. Sessions corresponded to teaching skills in ten trials by the participant. Each trial corresponded to teaching a different skill. For each trial, ten out of eleven DTT components could always be implemented by the participant (either C7 or C8 was not applicable depending on the learner’s response). Regarding the ten trials per session, the number of possible components was 100. If a given participant implemented 78 components correctly in a session, for example, the number of correct components would be divided by the number of possible components, 100, and the result would be multiplied by 100 to obtain a percentage. In this hypothetical case, the percentage would be 78%. A secondary DV of the research consisted of the cumulative number of correct and incorrect responses emitted by six children with ASD across four months of DTT teaching sessions following the end of BST. These children’s data corresponded to the last research condition concerning the maintenance of accurate teaching by the participants. This was the only stage in which the participants taught the skills specifically related to each child’s individualized ABA-based curriculum. Children’s data were also organized in graphs depicting simple frequencies of correct and incorrect responses by domain/type of repertoire taught.

### 2.5. Interobserver Agreement

Across many sessions involving most of the research conditions (baseline, BST with immediate and delayed feedback, assessment of generalization and maintenance), data collection related to the DTT components implemented correctly and incorrectly by the participants was conducted by an experimenter and an independent researcher. This was carried out to establish agreement rates between the two observers. For each of the six university participants (P1–P6) with whom IOA was obtained, agreement calculations were carried out by dividing the number of agreements by the number of agreements added to the number of disagreements and multiplied by 100 to obtain a percentage. The two observers showed agreement when they both recorded that a given participant either implemented DTT components correctly or incorrectly. For P1, IOA was obtained in 43.75% of the sessions and ranged between 96% and 100% (98% on average). For P2, IOA was obtained in 78.94% of the sessions and ranged between 90% and 100% (98% on average). For P3, IOA was obtained in 38.09% of the sessions and ranged between 94% and 100% (98% on average). For P4, IOA was obtained in 80% of the sessions and ranged between 44% and 100% (91% on average). For P5, IOA was obtained in 66.67% of the sessions and ranged between 93% and 100% (98% on average). For P6, IOA was obtained in 71.42% of the sessions and ranged between 82% and 100% (92% on average).

### 2.6. Experimental Design

The effects of the IV, BST, on the main DV, the number of DTT components implemented correctly by the university participants, were investigated with a single-case design consisting of nonconcurrent multiple baselines across participants (following recommendations by [10]). [43] ([43]) asserted that this design involved the observation of participants at different times. Data collection did not occur simultaneously. Baseline and intervention were set at different times for the participants.

In this study, six participants were allocated into two triads. Within each triad, the baseline condition was established first. Across sessions, each participant conducted sequences of ten trials to teach skills to a confederate. An experimenter collected data on the correct and incorrect implementation of DTT components. After a low and stable performance was demonstrated, an intervention started for one participant, and the others remained in the baseline condition. The intervention involved the four components of BST in the following order: didactic instructions about the skills to be taught, modeling (the experimenter rehearsed the correct implementation of trials with the confederate), role-play (each university student conducted trials with the confederate), and performance feedback provided by the experimenter per teaching trial across the sessions (immediate feedback).

Data on correct and incorrect implementation of DTT components were systematically collected during the role-play as in the baseline, but the experimenter programmed differential consequences for the participant’s performance. After an arbitrary learning criterion was demonstrated (two sessions with at least 90% components implemented correctly), another BST intervention condition was established. The only difference from the previous case was that performance feedback was delayed, that is, in each session, feedback was solely provided by the experimenter after the university participant administered the ten teaching trials to the confederate.

After the learning criterion was achieved, probe sessions were conducted to assess the generalization of teaching new skills to a child diagnosed with ASD instead of the confederate. Then, a maintenance (follow-up) condition was defined for the university participants over four months in which they had the opportunity to teach multiple skills to a child with ASD based on his/her individualized curriculum goals. Throughout this period, skill gains by the child were measured (secondary DV) and three probe sessions, to assess the participant’s maintenance of accurate skill teaching, were conducted. At follow-up, the child for each confederate remained the same. Thus, the research involved the following dyads concerning university participant and child: P1—CD1; P2—CD2; P3—CD3; P4—CD4; P5—CD5; P6—CD6. Finally, a self-assessment questionnaire was administered to obtain measures of social validation of the research procedures by the participant.

For each triad, when the effects of the IV on the main DV were demonstrated by the first participant (improvement in teaching accuracy through BST with immediate feedback), training was also established for the second university participant, who also went through the remaining research conditions thereafter. Finally, when the second participant also showed improvement in performance accuracy through BST with immediate feedback, that condition and the others in the study were also implemented with the third participant. Everything that was performed served the purpose of measuring experimental control, through the logic of the multiple baseline design, replicating the effects of interventions across different participants.

### 2.7. Ethical Procedures

This research was approved by the human research ethics committee of the Federal University of Maranhão (authorization 4.284.271). The six university students, the six children with ASD, and those responsible for them signed an informed consent form. The children themselves signed an assent consent to participate. Everything was in accordance with Resolution 510, of 7 April 2016, of the National Health Council in Brazil. All personal information was confidential. The participants (university students and children) could remove their consent or assent at any time, if they wished, without any harm. All participants had the right to know the rationale, objectives, and procedures to be used, with information on methods provided in clear and accessible language. Precautions to prevent any harm were made explicit. Participants were guaranteed confidentiality and privacy throughout the research process. The benefits of the research were made explicit. Participants were guaranteed access to the research results whenever they wished. They were also guaranteed reimbursement and forms of coverage for expenses arising from the research, if any. Participants received information about the ethics committee for research involving human beings, including telephone contact, which processed the research protocol. Participants also had the right to access the consent and assent record whenever they requested them.

### 2.8. Procedures

The study involved six conditions, which are described below. They are also summarized in a flowchart in Figure 1.

Baseline. All university participants had access to part of a manual ([29]), a week before data collection, with the purpose of reading descriptions on how to conduct DTT to teach the following ten skills: sitting still (e.g., the learner corrected their posture to an erect position while sitting and following the model provided by the interventionist); motor imitation (e.g., the learner clapped their hands based on the model provided by the interventionist); making requests (e.g., the learner said “can you give me juice?” in the presence of the interventionist); vocal imitation (e.g., the learner said “airplane” after the interventionist presented the verbal instruction “say airplane”); receptive identification of non-verbal stimuli (e.g., the learner pointed to the picture of a car in an array with several different pictures, and under the verbal instruction “point to the car” by the interventionist); making eye contact (e.g., the learner maintained eye contact with the interventionist for 10 s upon request); following instructions (e.g., the learner raised their hands after the interventionist provided the verbal instruction “put your hands up”); intraverbal (e.g., the learner said “apple” after the interventionist presented the verbal stimulus “what do you eat?”); labeling (e.g., the learner said “cat” when the interventionist showed a picture of cat and asked “what is this?”); receptive identification of non-verbal stimuli by function, feature, and class (e.g., the learner pointed to the picture of a tree in an array with several different pictures, and under the verbal instruction “show me something with leaves” by the interventionist).

When the baseline condition began, across sessions, each participant conducted ten DTT trials to teach the ten previously mentioned skills to a confederate, who pretended to be a learner with ASD. They followed scripts to randomly emit correct responses, incorrect responses, and no response across trials. In each session, each trial corresponded to the teaching of a different skill. During baseline, the university participants did not receive feedback from the experimenter regarding their performance accuracy to teach skills. The termination criterion for this research condition consisted of demonstrating a low and stable performance of correct implementation of DTT components across sessions.

BST with immediate feedback: During the first BST component, didactic instruction, classes on how to teach five of the previously mentioned skills in the baseline experiment were given. The classes were organized in PowerPoint slides. The contents explained how to provide DTT instructions, reinforce correct responses and correct errors (or no response during an allowed time) committed by a possible learner. The didactic instruction component was administered just once and lasted 30 min on average. Modeling was then conducted. The experimenter and the confederate, representing interventionist and learner, respectively, practiced ten DTT trials under a randomized order involving the five skills (two trials for each) for each participant to observe. The confederate followed scripts to emit correct responses, incorrect responses, and failure to respond across the trials, and the experimenter demonstrated the correct way to intervene on the confederate’s responses.

Thereafter, the role-play component was established, and each participant had to teach the five skills across sessions of ten trials (two trials for each skill) in a randomized order to the confederate, who followed scripts to emit responses the same way as in the modeling component. After each trial administered across sessions, the experimenter provided feedback on the participant’s performance in implementing the DTT components. For those implemented correctly, verbal praise was delivered. For those implemented incorrectly, corrective feedback was provided, that is, the experimenter clarified what was wrong and demonstrated the correct way to intervene with the confederate. The termination criterion for this research condition consisted of two sessions with at least 90% of DTT components implemented correctly.

It is important to emphasize that it was only during the role-play component that the experimenter collected data on the accurate teaching of skills (DTT components implemented correctly and incorrectly) by the university participants. In this research, the effects of each of the BST components were not investigated separately.

BST with delayed feedback: This research condition was similar to the previous one. The only difference was that, during the role-play component through which each university participant taught five skills to the confederate, performance feedback by the experimenter on correct and incorrect implementation of DTT was delayed. In other words, for each session, feedback was provided solely after the administration of ten teaching trials by each participant. The termination criterion was the same as the one from the previous research condition.

Generalization assessment: In this condition, as in the baseline condition, the university participants conducted ten DTT trials to teach skills along sessions, and no performance feedback was provided by the experimenter. As in the baseline condition, the participants taught the same ten skills, one per trial in each session. As was said before, five of these skills were involved in the training conditions through BST. The five remaining were never used in training. Their teaching by the participants during the current research condition served as a way of assessing the generalization of teaching integrity with new skills. In addition, across sessions, every participant taught a child diagnosed with ASD instead of a confederate. Therefore, the generalization of teaching integrity was also assessed through the teaching of six children with ASD (one per university participant).

Maintenance of accurate teaching and monitoring skill gains by children with ASD: Two weeks after the previous conditions, each university participant taught a child with ASD (the same one from the previous generalization assessment condition) once a week on average across up to four months. The skills taught were directly related to the curricular goals of each child regardless of the research. Children’s progress in skills was systematically measured during this condition. In addition, three follow-up probe sessions were conducted by the experimenter to assess the level of integrity of each university participant in teaching via DTT without performance feedback. The first session was conducted during the beginning of skills teaching for each child. The second session was conducted after the second month and the third and last one at the end. It is important to say that the period of the maintenance condition was four months because the psychology students, after BST in the university laboratory where data collection took place, collaborated as interns for at least a semester. After that, they could still collaborate, if they wished, as volunteers. It is also important to say that their training was not compulsory. They could remove their consent to participate at any time without any harm, if they wished.

Social validity assessment: The university participants rated the BST by answering a Likert scale questionnaire. It consisted of six questions with alternatives ranging from “1”, meaning “I completely disagree”, to “5”, meaning “I completely agree”. The following questions were presented to each participant: (1) I enjoyed participating in the BST; (2) I felt comfortable with the training process; (3) I learned important skills; (4) BST was effective for teaching children with ASD accurately; (5) I will continue to use the procedures I mastered to teach other skills to children with ASD; (6) I recommend the training to other interested people.

## 3. Results

Next, the percentages of DTT components implemented correctly by the six university participants (P1–P6) involved in the research conditions are presented. Figure 2 shows data from the first triad, that is, P1, P2, and P3.

As may be seen in Figure 2, at the end of the baseline, P1’s teaching integrity level consisted of 1% of DTT components implemented correctly (session S3). The BST condition with immediate feedback (BST IF) was terminated after three sessions with 96% teaching integrity (S6). The BST condition with delayed feedback (BST DF) was terminated after four sessions with 100% teaching integrity (S10). The generalization assessment condition ended with 98% teaching integrity (S13). Finally, by the end of the maintenance (follow up) assessment condition, teaching integrity level was 100% of DTT components implemented correctly (S16).

In the case of P2, baseline ended with 40% teaching integrity (S6). BST IF was completed in three sessions, highlighting 99% teaching integrity at the end (S9). BST DF was terminated in four sessions with 100% teaching integrity at the end (S13). Both generalization and maintenance conditions also finished with 100% teaching integrity (sessions S16 and S19, respectively). For P3, baseline ended with 1% teaching integrity (S9). Both BST IF and BST DF took three sessions to be terminated with 100% teaching integrity (sessions S12 and S15, respectively). By the end of both generalization and maintenance assessment conditions, teaching integrity was also 100% of DTT components implemented correctly (sessions S18 and S21, respectively). Figure 3 shows teaching integrity for each participant in the second triad (P4, P5, and P6) across the research conditions.

According to Figure 3, P4’s teaching integrity level by the end of the baseline was 25% of DTT components implemented correctly (S3). Both BST IF and BST DF ended after three sessions with 100% (S6) and 98% (S9) teaching integrity, respectively. Both generalization and maintenance (follow up) assessments were also terminated after three sessions with 100% teaching integrity (sessions S12 and S15, respectively). P5’s teaching integrity level by the end of the baseline was 44% (S6). Both BST IF and BST DF conditions ended in three sessions with 95% (S9) and 100% (S12) teaching integrity, respectively. Teaching integrity for both generalization and maintenance assessments were, at the end of three sessions, 91% (S15) and 100% (S18). P6’s teaching integrity level at the end of the baseline was 45%. Both BST IF and BST DF ended in three sessions with 99% (S12) and 100% (S15), respectively. Both generalization and maintenance assessment ended with teaching integrity levels of 96% (S18) and 100% (S21), respectively. Figure 4 shows the percentage of implementation errors committed by all university participants across the research conditions.

For the skills taught, each university participant could, within sessions involving ten trials, commit up to ten implementation errors of DTT components per trial. In a trial during which a confederate or child with ASD as learner did not emit a correct response, the implementation of a DTT component concerning reinforcer delivery was not possible. Likewise, in a trial during which the learner did emit a correct response, the implementation of a DTT component concerning a correction procedure was not possible. According to Figure 4, all participants committed many errors during baseline. However, when BST IF was established, implementation errors of DTT components were greatly reduced for everyone. In addition, implementation errors remained very low or absent in the following conditions of BST DF and GEN for all participants. Even more significant data were identified in the last condition, MAN, during which no participant made any errors in implementing DTT components across probes performed three times in a four-month period. It is important to remember that GEN and MAN conditions involved interactions to teach skills solely to children with ASD as learners. Figure 5 shows skill gains by the six participating children (CD1, CD2, CD3, CD4, CD5, and CD6) across these two mentioned research conditions (GEN and MAN).

As may be seen in Figure 5, CD1 accumulated 2047 correct responses and only 27 incorrect responses. Thirteen skills were taught during the MAN condition (from session three to session seventeen): 1—sitting still; 2—making eye contact; 3—answering questions; 4—describing non-verbal stimuli (pictures) through vocal verbal sentences; 5—retelling of stories; 6—three-step motor imitation; 7—three-step following instructions; 8—describing actions; 9—reading syllables; 10—copying numbers; 11—following the contours of syllables with the pencil; 12—receptively identifying pictures by function, feature, and class; 13—filling words in the blanks while singing.

CD2, in turn, accumulated 1145 correct responses and solely 97 incorrect responses. Ten skills were taught during MAN (from session three to session thirteen): 1—sitting still; 2—making eye contact; 3—performing dictation with words; 4—summing numbers; 5—subtracting numbers; 6—retelling of stories; 7—answering questions; 8—reading texts and answering comprehension questions; 9—filling words in the blanks while singing; 10—playing functionally.

CD3 accumulated 1104 correct responses and 62 incorrect responses. Thirteen skills were taught during MAN (from session three to session nine): 1—sitting still; 2—making eye contact; 3—motor imitation; 4—following instructions; 5—making vocal verbal requests; 6—describing actions vocally; 7—receptively identifying pictures by their names; 8—receptively identifying pictures by function, feature and class; 9—answering questions and other verbal stimuli; 10—vocal verbal imitation of words and sentences; 11—playing functionally; 12—drawing; 13—reading letters and numbers.

CD4 accumulated 1387 correct responses and 101 incorrect responses. Thirteen skills were taught during MAN (from session three to session 15): 1—sitting still; 2—making eye contact; 3—motor imitation; 4—following instructions; 5—receptively identifying pictures by function, feature and class; 6—labeling pictures; 7—playing functionally; 8—making requests through one-word utterances; 9—describing actions; 10—receptive identification of numbers; 11—receptive identification of letters; 12—answering questions and other verbal stimuli; 13—describing pictures through vocal verbal sentences.

CD5 accumulated 1576 correct responses and 90 incorrect responses. Seventeen skills were taught during MAN (from session three to session seventeen): 1—sitting still; 2—making eye contact; 3—three-step motor imitation; 4—following instructions; 5—making requests through sentences; 6—labeling actions in pictures; 7—answering questions and other verbal stimuli; 8—drawing; 9—playing functionally; 10—reading written or printed sentences; 11—reading numbers; 12—three-step following instructions with toys and other items; 13—performing dictation with words; 14—performing copy with words; 15—summing numbers; 16—subtracting numbers; 17—word reading.

CD6 accumulated 1348 correct responses and 81 incorrect responses. Sixteen skills were taught during MAN (from session three to session sixteen): 1—sitting still; 2—making eye contact; 3—three-step motor imitation; 4—following instructions; 5—making requests through sentences; 6—labeling actions in pictures; 7—answering questions and other verbal stimuli; 8—drawing; 9—playing functionally; 10—reading written or printed sentences; 11—reading numbers; 12—three-step following instructions with toys and other items; 13—performing dictation with words; 14—summing numbers; 15—subtracting numbers; 16—pairing arbitrary pictures. Next, children’s data are presented by domain/type of skill taught during the 4-month maintenance condition. Each of the following figures shows graphs depicting simple frequencies of correct and incorrect responses. Figure 6 shows data for CD1.

According to Figure 6, CD1 emitted (1) a total of 155 correct responses (99%) and 1 incorrect response (1%) in the sitting still program; (2) a total of 156 correct responses (99%) and 1 incorrect response (1%) in the eye contact program; (3) a total of 169 correct responses (99%) and 1 incorrect response (1%) in the retelling of stories program; (4) a total of 160 correct responses (98%) and 3 incorrect responses (2%) in the describing pictures through vocal verbal program; (5) a total of 165 correct responses (100%) and no incorrect response (0%) in the answering questions program; (6) a total of 166 correct responses (100%) and no incorrect response (0%) in the three-step motor imitation program; (7) a total of 166 correct responses (100%) and no incorrect response (0%) in the three-step following instructions program; (8) a total of 163 correct responses (100%) and no incorrect response (0%) in the describing actions program; (9) a total of 169 correct responses (100%) and no incorrect response (0%) in the reading syllables program; (10) a total of 170 correct responses (100%) and no incorrect response (0%) in the copying numbers program; (11) a total of 157 correct responses (98%) and 2 incorrect responses (2%) in the following the contours of syllables with the pencil program; (12) a total of 159 correct responses (96%) and 7 incorrect responses (4%) in the receptively identifying pictures by function, feature, and class program; (13) a total of 163 correct responses (100%) and no correct response (0%) in the filling words in the blanks while singing program. Figure 7 shows data for CD2.

As may be seen in Figure 7, CD2 emitted (1) a total of 114 correct responses (97%) and 3 incorrect responses (3%) in the sitting still program; (2) a total of 105 correct responses (90%) and 11 incorrect responses (10%) in the making eye contact program; (3) a total of 116 correct responses (93%) and 8 incorrect responses (7%) in the performing dictation with words program; (4) a total of 119 correct responses (93%) and 8 incorrect responses (7%) in the summing numbers program; (5) a total of 98 correct responses (77%) and 29 incorrect responses (23%) in the subtracting numbers program; 6) a total of 129 correct responses (99%) and 1 incorrect response (1%) in the retelling of stories program; (6) a total of 120 correct responses (92%) and 10 incorrect responses (8%) in the answering questions program; (7) a total of 109 correct responses (94%) and 7 incorrect responses (6%) in the reading texts and answering comprehension questions program; (8) a total of 108 correct responses (86%) and 17 incorrect responses (14%) in the filling words in the blanks while singing program; (9) a total of 130 correct responses (100%) and no incorrect response (0%) in the playing functionally program. Figure 8 shows data for CD3.

As may be seen in Figure 8, CD3 emitted (1) a total of 87 correct responses (97%) and 3 incorrect responses (3%) in the sitting still program; (2) a total of 81 correct responses (90%) and 9 incorrect responses (10%) in the making eye contact program; (3) a total of 90 correct responses (100%) and no incorrect response (0%) in the motor imitation program; (4) a total of 89 correct responses (99%) and 1 incorrect response (1%) in the following instructions program; (5) a total of 84 correct responses (93%) and 6 incorrect responses (7%) in the making vocal verbal requests program; (6) a total of 67 correct responses (74%) and 23 incorrect responses (26%) in the describing actions vocally program; (7) a total of 84 correct responses (95%) and 4 incorrect responses (5%) in the receptively identifying pictures by their names program; (8) a total of 82 correct responses (92%) and 7 incorrect responses (8%) in the receptively identifying pictures by function, feature, and class program; (9) a total of 90 correct responses (100%) and no incorrect response (0%) in the answering questions and other verbal stimuli program; (10) a total of 90 correct responses (100%) and no incorrect response (0%) in the vocal verbal imitation of words and sentences program; (11) a total of 90 correct responses (100%) and no incorrect response (0%) in the playing functionally program; (12) a total of 86 correct responses (98%) and 2 incorrect responses (2%) in the drawing program; (13) a total of 84 correct responses (93%) and 6 incorrect responses (7%) in the reading letters and numbers program. Figure 9 shows data for CD4.

Figure 9 shows that CD4 emitted (1) a total of 138 correct responses (93%) and 11 incorrect responses (7%) in the sitting still program; (2) a total of 145 correct responses (97%) and 4 incorrect responses (3%) in the making eye contact program; (3) a total of 130 correct responses (88%) and 18 incorrect responses (12%) in the motor imitation program; (4) a total of 121 correct responses (82%) and 27 incorrect responses (18%) in the following instructions program; (5) a total of 146 correct responses (97%) and 4 incorrect responses (3%) in the receptively identifying pictures by function, feature, and class program; (6) a total of 59 correct responses (98%) and 1 incorrect response (2%) in the labeling pictures program; (7) a total of 136 correct responses (97%) and 4 incorrect responses (3%) in the playing functionally program; (8) a total of 114 correct responses (93%) and 8 incorrect responses (7%) in the making requests through one) word utterances program; (9) a total of 81 correct responses (90%) and 9 incorrect responses (10%) in the describing actions program; (10) a total of 88 correct responses (98%) and 2 incorrect responses (2%) in the receptive identification of numbers program; (11) a total of 89 correct responses (99%) and 1 incorrect response (1%) in the receptive identification of letters program; (12) a total of 89 correct responses (99%) and 1 incorrect response (1%) in the answering questions and other verbal stimuli program; (13) a total of 82 correct responses (91%) and 8 incorrect responses (9%) in the describing pictures through vocal verbal sentences program. Figure 10 shows data for CD5.

Figure 10 shows that CD5 emitted (1) a total of 86 correct responses (97%) and 3 incorrect responses (3%) in the sitting still program; (2) a total of 113 incorrect responses (99%) and 1 incorrect response (1%) in the making eye contact program; (3) a total of 130 correct responses (100%) and no incorrect response (0%) in the three-step motor imitation program; (4) a total of 134 correct responses (98%) and 3 incorrect responses (2%) in the following instructions program; (5) a total of 127 correct responses (95%) and 7 incorrect responses (5%) in the making requests through sentences program; (6) a total of 143 correct responses (96%) and 6 incorrect responses (4%) in the labeling actions in pictures program; (7) a total of 121 correct responses (88%) and 16 incorrect responses (12%) in the answering questions and other verbal stimuli program; (8) a total of 127 correct responses (100%) and no incorrect response (0%) in the drawing program; 9) a total of 109 correct responses (91%) and 11 incorrect responses (9%) in the playing functionally program; (10) a total of 122 correct responses (96%) and 5 incorrect responses (4%) in the reading written or printed sentences program; (11) a total of 123 correct responses (86%) and 20 incorrect responses in the reading numbers program; (12) a total of 128 correct responses and 24 incorrect responses (14%) in the three-step following instructions with toys and other items program; (13) a total of 11 correct responses (92%) and 1 incorrect response (8%) in the performing dictation with words program; (14) a total of 90 correct responses (95%) and 5 incorrect responses (5%) in the performing copy with words program; (15) a total of 4 correct responses (40%) and 6 incorrect responses (60%) in the summing numbers program; (16) a total of 5 correct responses (50%) and 5 incorrect responses (50%) in the subtracting numbers program; (17) a total of 11 correct responses (92%) and 1 incorrect response (8%) in the word reading program. Figure 11 shows data for CD6.

Figure 11 shows that CD6 emitted (1) a total of 105 correct responses (97%) and 3 incorrect responses (3%) in the sitting still program; (2) a total of 107 correct responses (100%) and no incorrect response (0%) in the making eye contact program; (3) a total of 108 correct responses (100%) and no incorrect response (0%) in the three-step motor imitation program; (4) a total of 110 correct responses (100%) and no incorrect response (0%) in the following instructions program; (5) a total of 81 correct responses (90%) and 9 incorrect responses (10%) in the making requests through sentences program; (6) a total of 104 correct responses (94%) and 6 incorrect responses (6%) in the labeling actions in pictures program; (7) a total of 104 correct responses (84%) and 19 incorrect responses (16%) in the answering questions and other verbal stimuli program; (8) a total of 74 correct responses (96%) and 3 incorrect responses (4%) in the drawing program; (9) a total of 95 correct responses (98%) and 2 incorrect responses (2%) in the playing functionally program; (10) a total of 124 correct responses (98%) and 3 incorrect responses (2%) in the reading written or printed sentences program; (11) a total of 106 correct responses (97%) and 3 incorrect responses (3%) in the reading numbers program; (12) a total of 97 correct responses (98%) and 2 incorrect responses (2%) in the three-step following instructions with toys and other items program; (13) a total of 86 correct responses (86%) and 14 incorrect responses (14%) in the performing dictation with words program; (14) a total of 14 correct responses (93%) and 1 incorrect response (7%) in the summing numbers program; (15) a total of 9 correct responses (64%) and 5 incorrect responses (36%) in the subtracting numbers program; (16) a total of 10 correct responses (71%) and 4 incorrect responses (29%) in the pairing arbitrary pictures program. Next, Table 2 shows the results of the social validity assessment conducted with each of the six university participants who were trained through BST.

According to Table 2, overall, all university participants rated the six social validity questions with the highest score, that is, five. They were satisfied with the entire BST process.

## 4. Discussion

In this study, the components of BST were used to train six undergraduate psychology students in implementing the accurate teaching of multiple skills to a confederate pretending to be a child with ASD. BST was successful in the sense that all the university participants could teach five different skills to the confederate with a high teaching integrity level, that is, representing at least 90% of DTT components implemented correctly. Moreover, the high-performance accuracy generalized to the teaching of ten skills (including the five from BST with immediate and delayed feedback) to six children diagnosed with ASD.

These data corroborate the findings of previous studies on BST involving professionals, university students, and caregivers of children with ASD ([3]; [9]; [16]; [21]; [22]; [27]; [30], [31]; [33], [34]; [38]). However, the current investigation was also an extension because, by the end of the generalization assessment condition, all six university participants were able to accurately teach ten different skills as opposed to previous studies, whose participants taught fewer skills to learners.

This investigation also extended the previously mentioned studies on BST by training the university participants to accurately teach skills via DTT to children with ASD in the long term. During the follow-up/maintenance assessment condition, three probe sessions were conducted across four months and no implementation errors of DTT components occurred. During that period, the six university participants systematically taught multiple skills to the six children with ASD involved. The skills were directly related to the curriculum goals of each child, who attended the university laboratory once a week for 1 h and 30 min. Overall, the children accumulated more than 1000 correct responses across several sessions. These data may be considered meaningful because they reflect the skills gained by the children through their teaching by university students who were specifically trained via BST for this purpose. Before the onset of the study, the university students had no experience with ABA applied to ASD.

Therefore, recovering the three guiding questions of the current research, it is considered that the hypotheses were confirmed. In other words, (1) BST effectively trained six university students to accurately teach multiple skills via DTT to a confederate pretending to be a child with ASD; (2) BST generalized the accurate teaching of multiple skills, including new ones not covered in training, to children with ASD across post-training probes; (3) the university students’ high teaching integrity during the provision of DTT and the children’s acquisition of skills, according to their individualized curriculums, were demonstrated across four months on average. This outcome adds to the previously mentioned literature, which suggests BST is a possible gold standard to teach individuals to implement procedures such as DTT to learners with ASD. Moreover, previous research has discussed that BST is also effective for different populations and behaviors. In a recent meta-analysis by [18] ([18]), for example, the authors examined studies designed to train people to teach others and to directly teach skills to individuals. Some studies focused on establishing conversational skills for college students with learning disabilities, athletic skills for young people (e.g., soccer, skateboarding skills), safety skills for children (e.g., lockdown drill responses, fire safety, abduction prevention, and reporting unsafe packages), and computer skills for adolescents and adults with ASD. Other investigations aimed at peer-initiated BST to implement DTT and conversational skills for learners with atypical development. Some studies involved training school staff, researchers, and parents to implement BST. All identified investigations employed single-case designs, and the methods by Flowers et al. to examine the effectiveness of BST (percentage of nonoverlapping data, percentage of all overlapping data, and nonoverlap of all pairs) revealed through studies that BST was effective.

Regarding the recent meta-analysis conducted by [17] ([17]), which was more specifically concerned with identifying single-case studies to assess the effects of BST on the implementation of DTT with accuracy, the current research was able to address some limitations. First, unlike the studies on BST from the mentioned meta-analysis, the current investigation was able to systematically include consistent information on data from six children with ASD as skill learners through DTT, and this was carried out across four months on average. It is important to not only assess the potential of BST in increasing the levels of teaching integrity by those interested in implementing ABA procedures but also to measure the potential of BST to influence important developmental gains for children with ASD, whom the interested people have been trained to teach. The skills taught to the children during that period, after BST and the assessment of the generalization of accurate teaching, were aligned with their individualized curriculum goals. However, one limitation was the fact that, during this study, pre- and post-test measures of skill gains were not obtained through standardized screening protocols.

It is important that potential future replication studies address this limitation. Furthermore, a very important limitation also needs to be pointed out. Many programs taught to the children during the last four months were at a maintenance level. That means that the children already had many skills, which represents a methodological inaccuracy. These children were enrolled in the university laboratory where they had been receiving ABA interventions on DTT for many months regardless of the study. They had progressed in many skills over time, before the onset of the research. Currently, after this research, some children have even been discharged from laboratory activities. Ideally, new children, with whom a careful pre-test to identify skill deficits could be applied, should have been defined as learners in this investigation. However, there were no openings available for new children at the university laboratory during the period of the research. It was decided to select the most skilled children from the laboratory, who also did not exhibit significant levels of disruptive/interfering behaviors. Therefore, all participating children were quite cooperative, which certainly also influenced good performance in DTT from the very beginning.

This study also addressed another limitation from the previous literature. In the meta-analysis by [17] ([17]), out of a total of 224 participants, only 51 (23%) were subjected to a maintenance phase. In contrast, all participants in the current research went through a maintenance phase. This measure is important to assess if BST produces long-lasting effects on teaching integrity, which was the case in this study. Still regarding maintenance, the studies identified in Fingerhut and Moyaert’s meta-analysis revealed that, unlike the current investigation, after BST was finished, the level of integrity in the implementation of DTT was slightly reduced during maintenance conditions.

The current study was carried out in a research laboratory from a Brazilian University. Although behavior analysis in Brazil is a well-established area of knowledge production in basic research, efforts in producing experimental studies of applied importance, involving single-case designs and ASD, are still new and show limitations that need to be addressed. A recent systematic review investigation conducted by [13] ([13]) resulted in the identification of 59 research articles published in Brazilian and international journals from 2007 to 2024 that investigated the effectiveness of single-case methodology on the modification of socially important behaviors of individuals with ASD. [13] ([13]) also assessed the quality of the studies based on the indicators of the “What Works Clearinghouse” ([44]). It was noticed that, out of 59 articles, only 14 fully followed the WWC standards.

However, [13] ([13]) emphasized that this did not imply that the 45 studies that did not follow the standards were not relevant. Forty-three studies were published in Portuguese whereas sixteen were published in English. A growing trend was observed in publications, especially from 2015 onwards. The authors discussed that publications in Portuguese, which represented most of the articles, benefited many professionals and other interested people who delivered ABA-based services and who were not very proficient in the English language. Nevertheless, it was discussed that articles in Portuguese may limit access to researchers outside Brazil, and that these studies may not be included in international reports on ASD research and the effectiveness of ABA-based interventions on a global level. The Brazilian studies from the systematic investigation demonstrated a greater concern with procedures to teach academic and language skills to learners with ASD who demand less support.

It was discussed that more investigations were warranted on more prevalent behaviors in individuals with ASD, such as interfering behaviors, communication skills at a more basic level, and functional behaviors related to independence in the community. It was also noticed that there was a lack of research aimed at designing behavioral interventions for adolescents and adults. In this sense, due to the lack of studies for this population, it was discussed that as people with ASD aged, the existence of evidence-based practices became limited or scarce. The authors also said that new research needed to include social validity methods to precisely measure the acceptance level of interventions and results. According to [13] ([13]), social validity allows one to know if interventions are based on meaningful goals and acceptable strategies and involve positive outcomes for those who are the target of the interventions and stakeholders. Therefore, De Souza et al. indicate new avenues for studies involving single-case designs in Brazil based on the profiles of learners with ASD who have not been subject of research. It is reasonable to assume that this may also include a call for studies which may investigate the effectiveness of BST in training professionals, paraprofessionals, and caregivers in providing accurate ABA-based interventions. The interventions may refer to DTT with the purpose of teaching skills to learners with ASD, especially those who, for example, demand the teaching of language and communication skills at a more basic level. Less attention has been given to this portion of the population in research involving BST and DTT, justifying the importance of new studies.

As was previously said, the current research assessed the effectiveness of BST in training university students to teach multiple skills through DTT to confederates and children with ASD. A non-concurrent multiple-baseline design, a type of single-case design, was used to determine the effectiveness of BST components, which was confirmed. This study followed the standards for single-subject studies from [44] ([44]), seeking to address a problem indicated in the systematic review by [13] ([13]), in the sense that many Brazilian studies from the review did not follow the WWC standards. In other words, in the current investigation, data were presented in graph and table format; the independent variable (IV–BST) was systematically manipulated; interobserver agreement (IOA) was obtained by two observers across all experimental conditions and was determined to be above 20% of the sessions per condition; the IOA level between the observers was determined to be above 80% across the experimental conditions; it used a three-tier multiple-baseline design for each of the two triads of university students who participated; each of the study’s phases involved at least three data points with little or no variability; also, the research involved the assessment of generalization, maintenance, and social validity of procedures and results. Everything was conducted with the aim of ensuring the internal and external validity of procedures and results, which is expected from single-case studies.

All the skills taught to the six children with ASD in the study over the course of months tended to have a positive impact on their development, possibly influencing them to be more functional in different environments. This may be a prerequisite for the development of more spontaneous and naturalistic communication and greater independence in carrying out daily routines, including engagement in behavioral sequences that are important for promoting health and preventing infectious diseases such as COVID-19 ([11]).

In addition, as said before, the three probe sessions conducted by an experimenter across four months revealed that the university participants implemented all possible DTT components without any errors. All these data possibly represent the most relevant contributions of this study, as they suggest that BST produced a high impact on skill gains for all the 12 participants (children and university students) in the long term. In addition, all university participants rated their training across the six questions of the social validity assessment with the highest score.

The results of this study corroborate the data involving caregivers from research by [20] ([20]). In that study, the caregivers systematically taught at least ten different skills to 6 out of 17 children with ASD submitted to intensive behavioral interventions over a year. It was a longitudinal study, which did not involve a single-case methodology. Post-tests involving a standardized developmental assessment protocol showed that the six mentioned children demonstrated significant gains in skills compared to the others to whom multiple skills were not targeted. In the current study, all six participating children with ASD showed significant skill gains across multiple targets as well. These targets consisted of ten different skills across most of the research conditions until generalization assessment, and several other skills across the sessions in the follow-up/maintenance assessment condition. However, as a limitation of this study, no standardized developmental assessment tool was used unlike in the study by Gomes et al. with caregivers.

Another limitation of this study refers to the number of sessions in which the six children with ASD were submitted to DTT interventions across the four months of the maintenance assessment condition. Each child was taught only one day per week for 1 h and 30 min. It is expected that many children with ASD, with difficulties in developing multiple skills, be submitted to more intensive interventions lasting several hours per week. However, the children in this research came from low-income families, unable to afford the costs of more intensive weekly interventions. The university laboratory in which they were taught was only able to allow one session per week, so more children could be involved. Even so, it should be noted that, as another limitation of the research, some children sometimes missed sessions due to variables such as illness, or even difficulties experienced by their caregivers due to work and other competing activities.

As was said before, an alternative may be training parents and other caregivers to implement interventions themselves ([23]), but it may be a challenge to train caregivers in a country where many people with low income need to spend many hours at work or on household chores, often being unavailable ([3]; [20]). Then, we suggest that joining efforts to train an ever-increasing number of university students as future professionals, through BST, represents a way of increasing the number of interventionists who are better qualified to provide ABA-based services with quality and methodological precision, reducing costs for families ([31]).

In this research, the effects of the four BST components were not assessed separately, making it impossible to isolate the effects of each component on teaching integrity levels by the university participants. However, as said before, the meta-analysis research developed by [17] ([17]) discussed that the effects of BST components were better when they were administered together. These authors even stated that when the four components of BST were employed together, they were statistically significant. The current study took this argument into consideration and assessed the effects of the BST components together.

Other research on training university students and caregivers also discussed possible alternatives to in-person BST. [6] ([6]) evaluated the effectiveness of an instructional video modeling approach to train mothers to implement DTT to teach skills to their children with ASD. Although access to instructional video modeling increased the mother’s levels of teaching integrity, the authors discussed that the instructional video modeling was more effective as part of a more comprehensive training program involving interactions with the behavior analyst trainer. [19] ([19]) assessed the effectiveness of a self-instruction manual to train university students to accurately teach skills via DTT to children with ASD. Although the manual improved the participants’ performance, the authors also discussed that the self-instruction manual should be part of a more comprehensive approach involving other strategies such as, for example, performance feedback by the trainer.

This discussion strengthens the hypothesis that BST, with its four components, seems to be a gold standard in training people interested in implementing ABA-based interventions, such as DTT, appropriately. However, the delivery of in-person BST may pose some important limitations. People interested in learning how to implement proper ABA interventions may encounter limitations for an in-person BST process to conduct DTT due to variables such as, for example, living in a rural area with little access to training and supervision services, and health-related variables such as, for example, a pandemic like COVID-19, establishing the need for social isolation ([8]; [14], [15]; [35]; [41]).

Recently, the feasibility of remote BST, as an alternative to in-person training, has been discussed in the literature. This may be an important measure to increase the likelihood of training professionals, parents, and other caregivers to implement ABA procedures such as DTT. It has already been said that parent training also represents EBP ([24]; [36]; [40]) and helps to increase the intensity of interventions for children with ASD and reduce financial costs for families. Recent studies have demonstrated the effectiveness of remote BST using computers or laptops and Internet connections (telehealth services) to train parents on the accurate provision of DTT to their children with ASD ([2]; [3]; [22]). In the systematic review study by [17] ([17]), regarding the 46 studies on BST to implement DTT accurately, only 13% of the participants were parents, and the authors recommended more research to determine whether BST would benefit parents with less experience in implementing DTT with their children.

[2] ([2]) used remote BST to train the parents (father and mother) of a child suspected of possibly having an ASD diagnosis, since he showed deficits in motor imitation, pointing, and making requests vocally. Through remote BST, the parents learned how to accurately teach skills via DTT as well as under more naturalistic circumstances regarding daily routines such as taking a shower, having meals, and functional play. In these cases, the skills were evoked according to the immediate interests of the child. Improvements in the integrity of the intervention delivery by the parents were demonstrated, and it was discussed that they possibly helped to mitigate ASD symptoms. As limitations, the authors pointed out the involvement of only one child in the research, and the fact that the parents demonstrated high levels of teaching integrity during the baseline, which may have obscured the possible causal relationship between parental training and improvements in the integrity of repertoire teaching for the child with ASD.

[3] ([3]) also trained parents (father and mother) through remote BST to teach several skills via DTT to their child diagnosed with ASD. Each parent learned how to accurately teach two academic and two communication skills. However, a limitation of the study was that no direct assessment was conducted with the child to establish his individual needs. Moreover, his skill gains were not consistently measured, which also represented a limitation. Generalization of accurate teaching integrity by the parents was demonstrated in a different room in the residence, which consisted in an environment with more distractions, representing more naturalistic circumstances. In addition, the maintenance of high teaching integrity was also demonstrated a week after training was finished. The parents rated their training positively and stated that the child learned important skills through them.

[22] ([22]) trained three mothers through remote BST to teach the skill of labeling pictures via DTT to their children with ASD at home. Remote BST was successful, and, by generalization, the mothers accurately taught the children to label new pictures. There was evidence of the maintenance of high teaching integrity a month after training was finished. The mothers rated their training positively and recommended it to more people who might be interested in the training process. The children demonstrated the acquisition of labeling skills, and this was consistently measured along the research phases. Previously, the authors had conducted direct assessments with the children to establish their individual needs. In this study, however, training the mothers to teach only one type of skill was a limitation.

In Brazil, although remote BST with Internet connection may be a viable option to train parents and caregivers of children with ASD, as well as university students and professionals with limitations for an in-person training process, many people, especially those living in the Northeast region of the country, do not own a computer or laptop. In this sense, a possible alternative to this may be the use of devices such as smartphones, which many people, regardless of their socioeconomic status, usually have. New research on remote BST is warranted to investigate the feasibility and effectiveness of training using this type of device. This should be established for parents, other caregivers, university students, professionals, and paraprofessionals who might benefit from the proposal and implement ABA-based interventions, such as DTT, accurately and at a lower cost without having to go to training locations that may be difficult to access. In the study by [20] ([20]) previously discussed, for example, 17 family caregivers were subjected to in-person training to systematically teach their children with ASD several skills over a year. As a matter of fact, initially, the researchers intended to train 55 family members. Nevertheless, most of them were financially unable to attend the meetings because either they could not afford transportation to the location, or they had no one who could take care of their children while they went to training.

The current study originally intended to involve parents of children with ASD as participants as well, but during the recruitment period, no one had time available for in-person training. Thus, this study only involved psychology university students who showed interest in becoming professionals in the future to provide effective behavioral interventions to help improve the quality of life of children with ASD. It was assumed that training future professionals also represented an important way of reducing costs and facilitating access to specialized ABA services for a greater number of learners and their families. Moreover, this research systematically measured and informed data regarding gains in skills for the children with ASD involved, as opposed to most studies in the meta-analysis by [17] ([17]), which did not do this consistently. Furthermore, the maintenance condition showed that the effects of BST on high teaching integrity levels of the participating university students were long-lasting and errorless, as opposed to participants from the studies in the mentioned meta-analysis, who showed a slight reduction in their teaching integrity. The limitations in studies from the systematic review by [13] ([13]) warrant more research to produce and refine the evidence of effective ABA-based interventions to induce relevant behavior change, as well as new research on BST to train interested people to provide interventions, including DTT.

Although the benefits of the four BST components in training people interested in promoting socially relevant behavior change in learners with ASD are well established in the literature, more research to assess the generality of the effectiveness of BST on training people to implement DTT accurately is warranted. According to [12] ([12]), there is still limited research on the use of BST in learners with ASD who are minimally vocal or who show significant delays in skills. Likewise, the meta-analysis by [17] ([17]) also confirmed the need for more research with this population. The systematic review, regarding research in Brazil on single-case designs and ASD by [13] ([13]), also revealed that studies involving less skilled, minimally vocal or non-vocal learners were still scarce in the mentioned country, despite being a country in which research on behavior analysis had been conducted for decades. In the current study, although the learners with ASD were not minimally vocal or did not show significant delays, data on their skill acquisition were consistently taken and informed, which addressed a limitation pointed out in the meta-analysis by Fingerhut and Moeyaert. Moreover, in the current investigation, the university students who participated learned to teach the children involved multiple skills accurately through DTT, at least ten, following recommendations by [20] ([20]). Previous studies on BST involved teaching fewer skills. It is important that future studies assess the effects of BST on the possible accurate teaching of multiple skills to minimally vocal learners with ASD.

In this study, the university participants were trained to teach skills through DTT only in the context of the distraction-free environment of a university laboratory research room. However, it is important that future studies also assess the generality of training and the teaching of multiple skills in more natural environments, such as the participants’ residence, which may more accurately reflect the learners’ day-to-day lives. In this sense, it is important that the learners exhibit characteristics associated with a more progressive approach according to [26] ([26]). In other words, for example, learners in more natural environments should demonstrate more accurate attending behaviors and more complex language skills, which tend to be prerequisites for the use of more wordy and complex instructions to teach them and for the use of varied topographies of instructions and less intrusive prompt types and prompting systems.

Another limitation of this study refers to the assessment of social validity. Although this measure was consistently obtained with the university students regarding their training, that is, all of them rated the effectiveness of BST highly, social validity measures by the children’s parents or other caregivers, regarding skill development by the children, were not consistently obtained. Nevertheless, the parents and other caregivers of the participating children with ASD in this investigation informally said that the children showed improvements that reflected their functioning in their day-to-day environments. In other words, the caregivers said they were showing better attending behaviors and communicating at home and at school. In any case, it is important that future studies on BST aiming to teach multiple skills also consistently collect data on the social validity of the results representing skill gains in the children’s repertoire. This, in fact, represents one of the important parameters for validating research in ABA to ASD involving single-case designs ([44]).

Still regarding social validity, it was said that the university students rated their training solely through a questionnaire with objective questions in the Likert format. However, it is important that future studies also allow university participants to give their perspectives on the BST and implementation of DTT with comments for a more qualitative interpretation. Likewise, future studies may also obtain qualitative feedback from participating children with ASD, which did not happen in the current research.

Although the current study identified and sought to address gaps identified in previous research on BST for DTT implementation ([17]), ABA applied to ASD has been the target of criticisms by some communities, that is, some autism rights and neurodiversity activists. According to [25] ([25]), there are some concerns that underlie the criticisms, such as, for example, concerns about Ivar Lovaas and his UCLA Young Autism Project, which improved the quality of life of those with ASD, but in the past, it used moderate punishment strategies for a brief period to decrease interfering behaviors. However, practices based on the use of positive reinforcement have always been defined as a priority. Another concern refers to the intensity of interventions. Some people may advocate, for example, that 40 h of intervention per week is an exaggeration. However, Leaf et al. advocates that the number of hours per week must be based upon individual needs, and that there is no research to support that any intervention intensity may be harmful for learners.

Considering the research from this manuscript, the participating children at the university laboratory, where data collection took place, only attended the laboratory once a week for 1 h and 30 min, regardless of the study. This happened so the university could provide ABA services for a higher number of children, whose parents and caregivers were unable to pay high prices for these services. Regardless, most of the children developed several important skills, which helped them to be more functional and independent. It has also been suggested by some studies that ABA may lead to negative outcomes such as depression, post-traumatic stress disorder (PTSD), and anxiety. However, [25] ([25]) pointed out that these studies were based on reports, which lacked reliability or replication. Despite criticism by some groups of individuals, ABA has been producing scientific evidence of the effectiveness of its practices for many years, which also includes, for example, BST and DTT. It has also been discussed that the continued examination of its applications to solve significant human issues is warranted to refine or replace them with new and better ones.

## 5. Conclusions

BST was successfully conducted on six psychology undergraduate students to accurately teach multiple skills via DTT to a confederate and six children diagnosed with autism in the context of a university laboratory. The follow-up/maintenance condition over four months on average indicated that BST effects were long-lasting. The university participants implemented all possible DTT components in three probe sessions without errors, and the children demonstrated over 1000 correct responses across multiple skills, although each of them was only taught one day per week for 1 h and 30 min. An avenue for future research may be replicating the procedures of the current study with parents and other caregivers as participants. One possible way to increase the likelihood that caregivers from a country such as Brazil with so many socio-economic inequalities participate in training may be through Internet with remote meetings. The meetings may be held with devices such as smartphones, which many people own, regardless of their socio-economic status. Caregivers and trainers would not need to be physically in the same place. All four BST components may be administered remotely. The trainers may record didactic audio and video instructions on multiple skills that caregivers would teach to their children with ASD. During the modeling component, an experimenter may record videos of two actors representing the roles of the interventionist and child with ASD rehearsing DTT. Caregivers could download the videos to watch whenever they need. During the role-play component, caregivers would conduct DTT directly with their children with ASD, and the experimenter, after seeing the process through videoconferencing in his/her smartphone, would provide performance feedback to increase the caregivers’ accuracy in teaching their children. As performance improves, generalization and maintenance assessment conditions in the long term may be planned and established. The authors of the current study hope that this possibility of remote investigation can help establish multiple skills in children with ASD and that skill gains can be systematically measured over several months. It is also hoped that minimally vocal and less skilled children are also included as learners, as recommended by recent meta-analysis investigations. Moreover, since it has also been discussed that the components of BST are more effective when used together, a separate component analysis may not be necessary.

## Figures and Tables

**Figure 1 behavsci-15-00742-f001:**
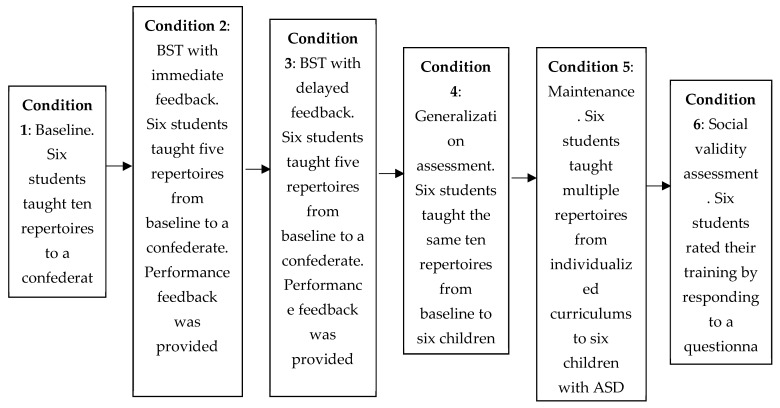
Flowchart of the research conditions.

**Figure 2 behavsci-15-00742-f002:**
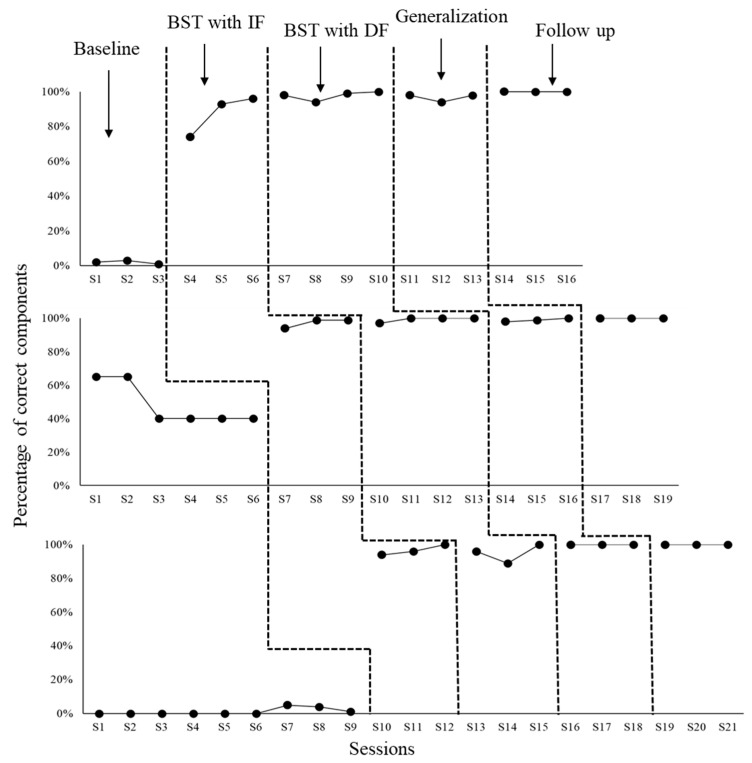
Percentages of components implemented correctly by P1, P2, and P3 at different stages. Note: The upper, middle, and lower graphs represent data from P1, P2, and P3, respectively. Each graph shows the percentage of DTT components implemented correctly during the teaching of skills to a confederate and/or child with ASD at different stages of the study (baseline, BST with immediate feedback—IF, BST with delayed feedback—DF, generalization assessment and follow up/maintenance assessment). Interactions involving children with ASD as learners only occurred during the stages designed to assess the generalization and maintenance of teaching integrity via DTT.

**Figure 3 behavsci-15-00742-f003:**
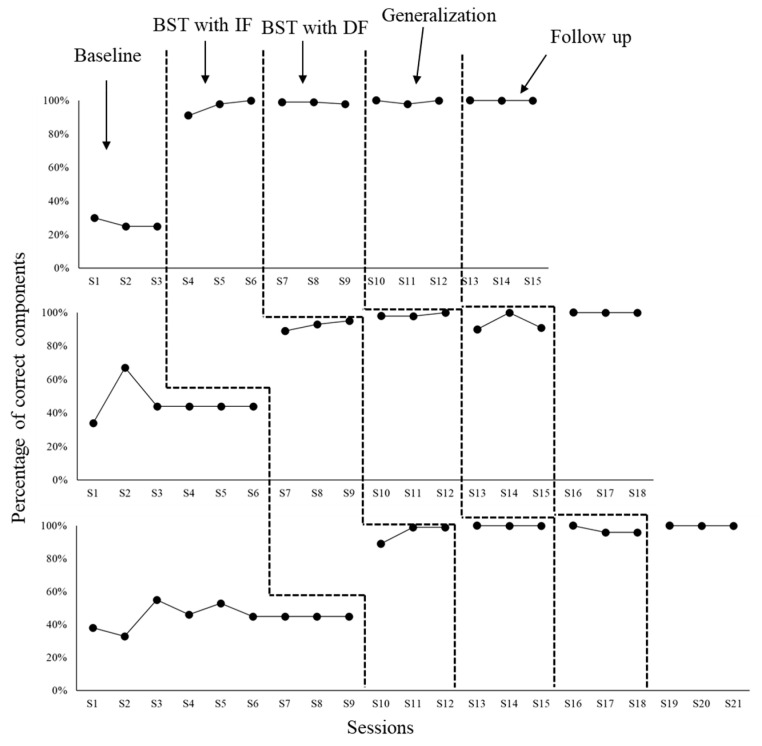
Percentages of components implemented correctly by P4, P5, and P6 at different stages. Note: The upper, middle and lower graphs represent data from P4, P5, and P6, respectively. Each graph shows the percentage of DTT components implemented correctly during the teaching of skills to confederate and/or child with ASD at different stages of the study (baseline, BST with immediate feedback—IF, BST with delayed feedback—DF, generalization assessment and follow up/maintenance assessment). Interactions involving children with ASD as learners only occurred during the stages designed to assess the generalization and maintenance of teaching integrity via DTT.

**Figure 4 behavsci-15-00742-f004:**
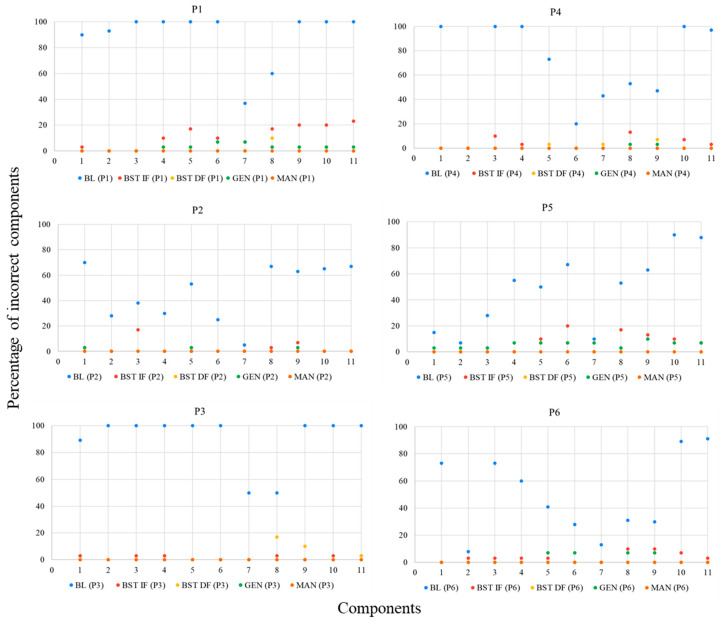
Percentage of implementation errors of DTT components by each participant. Note: Each DTT component (from C1 to C11) was described in the method section on data collection instruments. Each dispersion graph shows the data from one of the six university participants (P1–P6). The data from the graphs refer to each of the research conditions, that is, baseline (BL), BST with immediate feedback (BST IF), BST with delayed feedback (BST DF), generalization assessment (GEN), and maintenance assessment (MAN).

**Figure 5 behavsci-15-00742-f005:**
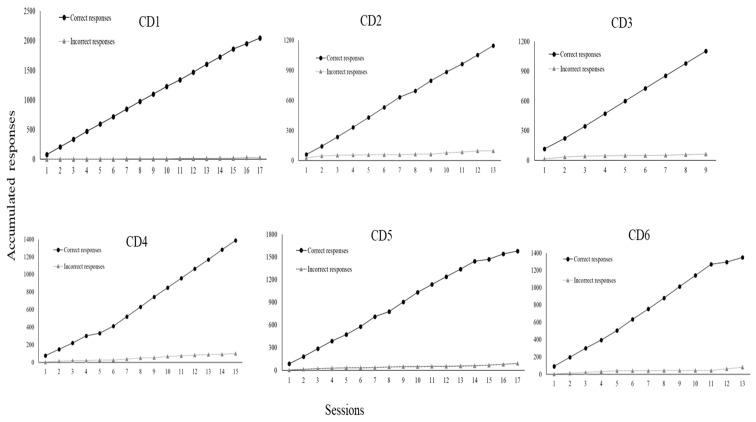
Repertoire gains by the participating children with ASD. Note: Each graph shows accumulated correct and incorrect responses across sessions by the six participating children with ASD as learners (from CD1 to CD6). The first three sessions with each child refer to skill acquisition during GEN condition. The skills were the same ten assessed in baseline during interactions with a confederate pretending to be a child with ASD. The remaining data refer to skills considering the curricular goals for each child regardless of the study. Some skills were different than those involved in the previous research conditions.

**Figure 6 behavsci-15-00742-f006:**
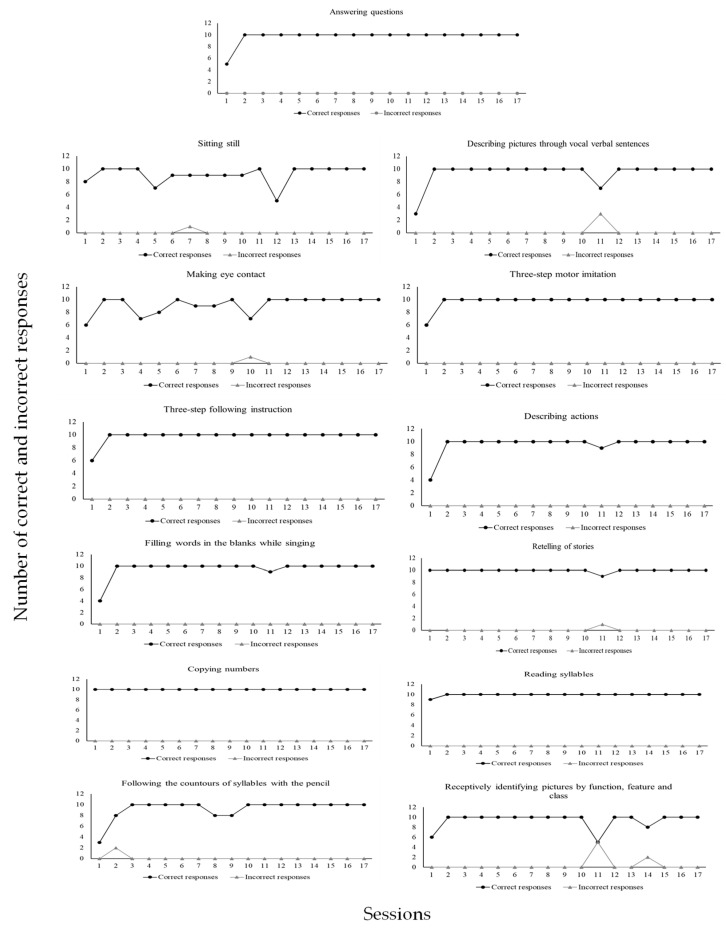
Skill gained by child CD1 by domain/type of skill taught during maintenance condition.

**Figure 7 behavsci-15-00742-f007:**
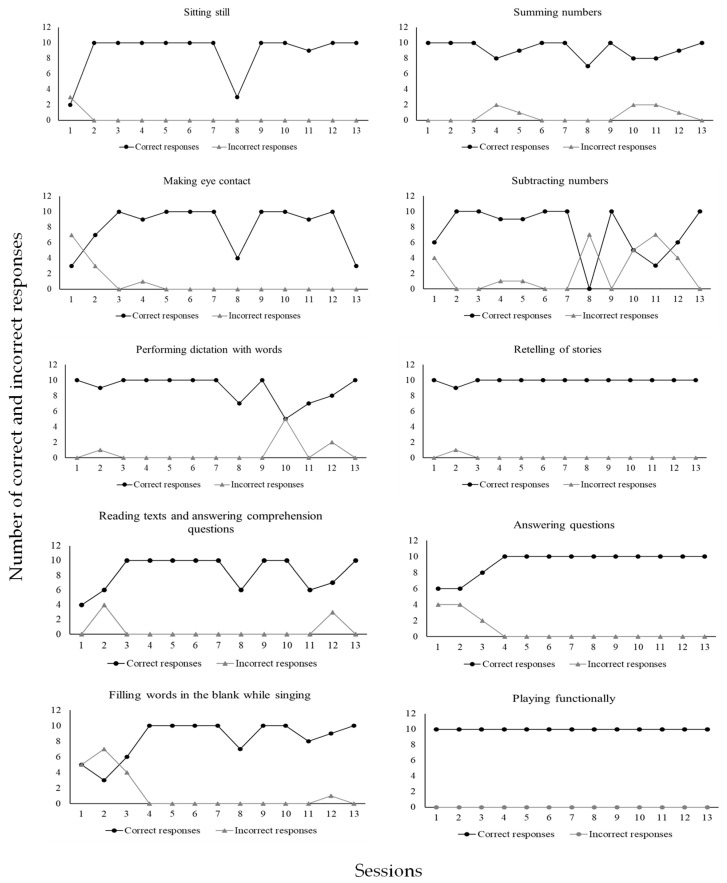
Skill gained by child CD2 by domain/type of skill taught during maintenance condition.

**Figure 8 behavsci-15-00742-f008:**
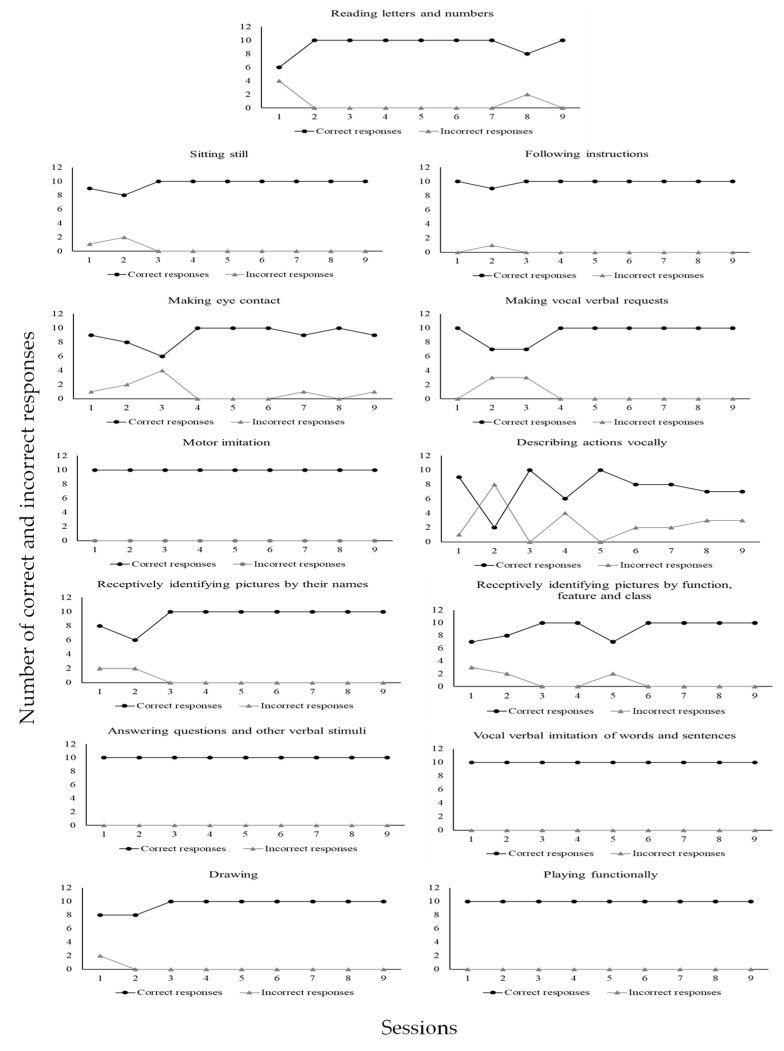
Skill gained by child CD3 by domain/type of skill taught during maintenance condition.

**Figure 9 behavsci-15-00742-f009:**
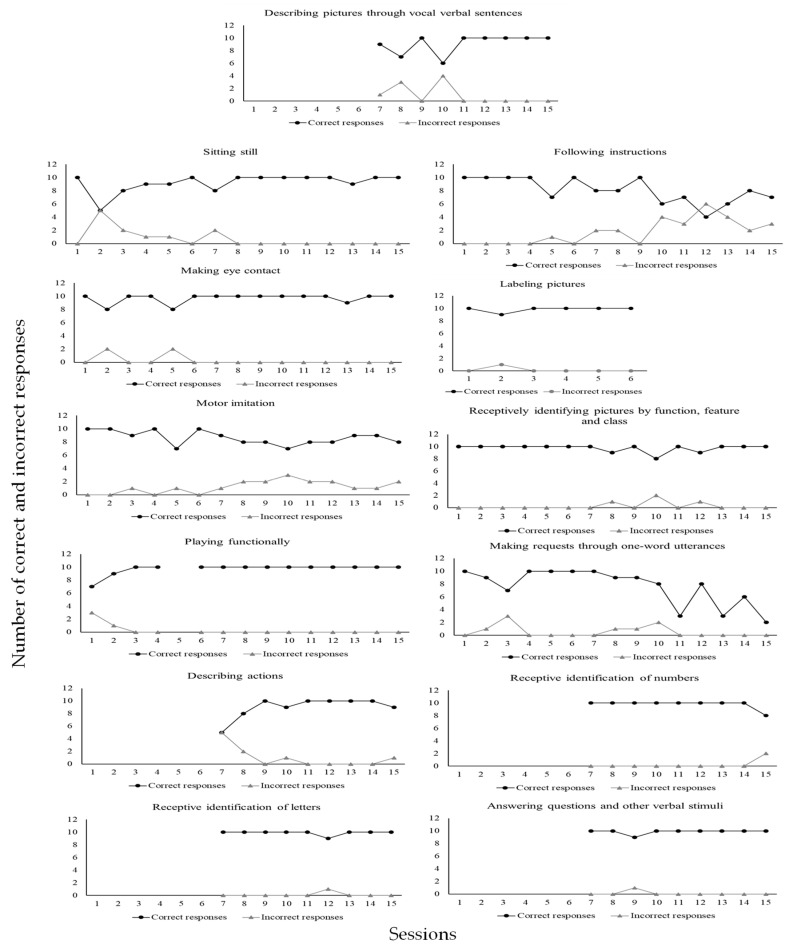
Skill gained by child CD4 by domain/type of skill taught during maintenance condition.

**Figure 10 behavsci-15-00742-f010:**
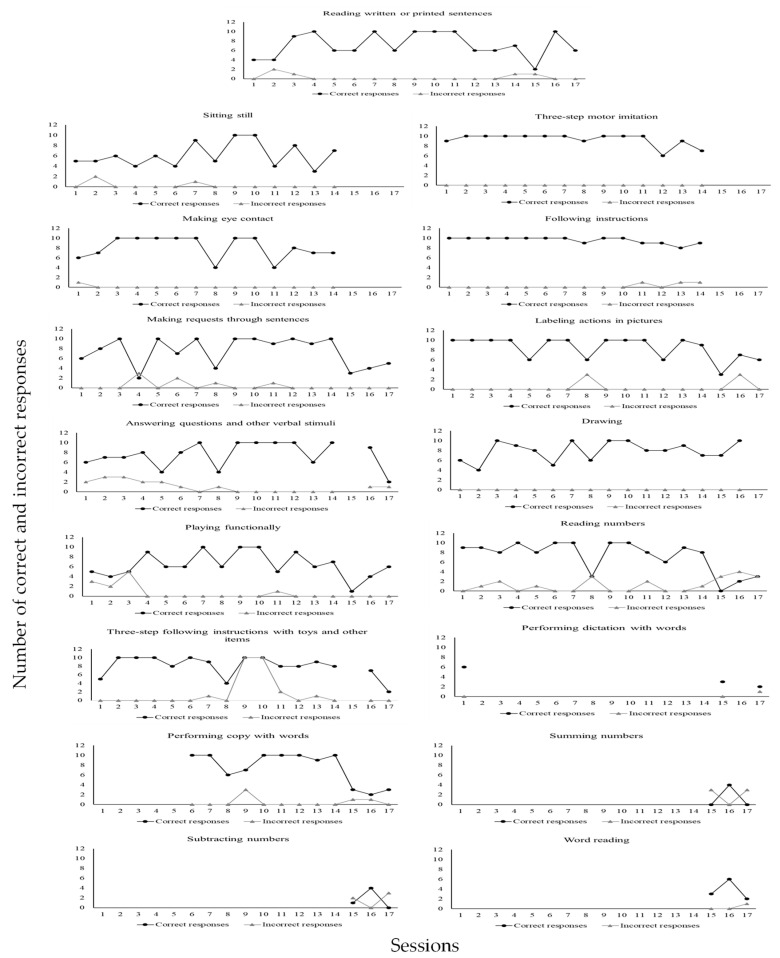
Skill gained by child CD5 by domain/type of skill taught during maintenance condition.

**Figure 11 behavsci-15-00742-f011:**
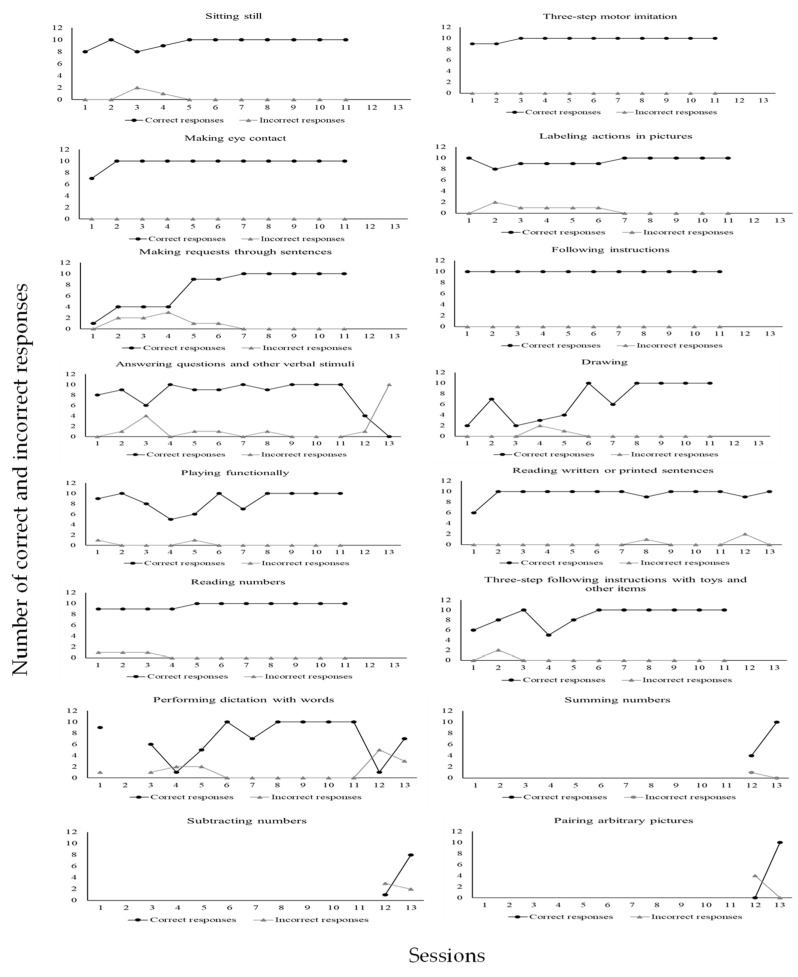
Skill gained by child CD6 by domain/type of skill taught during maintenance condition.

**Table 1 behavsci-15-00742-t001:** 11 DTT components of the research.

DTT Components
C1—The participant kept the area free from distractions, removing unnecessary items from the table surface
C2—The participant organized necessary materials such as pencil, recording sheet, stimuli cards and reinforcers to be within reach
C3—The participant evoked behaviors that indicate attention such as eye contact and shoulders facing the confederate/learner
C4—The participant waited for the emission of behaviors that indicated sustained attention for at least 2 s
C5—The participant provided well-articulated and clear verbal instructions in a way that they did not use more than four words
C6—The participant waited for a response from the confederate/learner for up to 5 s
C7—If the confederate/learner emitted a correct response, the participant provided praise and access to an arbitrary reinforcer
C8—If the learner emitted an incorrect response, the participant implemented a correction procedure
C9—The participant recorded the learner’s response
C10—The participant removed used stimuli before the provision of the next trial
C11—The participant waited 5 s before starting the next trial

**Table 2 behavsci-15-00742-t002:** Social validity assessment results.

1. I enjoyed participating in the BST.	P1	P2	P3	P4	P5	P6
2. I felt comfortable with the training process.	5	5	5	5	5	5
3. I learned important skills.	5	5	5	5	5	5
4. BST was effective for teaching children with ASD accurately.	5	5	5	5	5	5
5. I will continue to use the procedures I mastered to teach other skills to children with ASD.	5	5	5	5	5	5
6. I recommend the training to other interested people.	5	5	5	5	5	5

## Data Availability

The authors make the raw data underlying the conclusions in this article available upon request.

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
