# Peer review of "Training University Psychology Students to Teach Multiple Skills to Children with Autism Spectrum Disorder"

_behavsci, 2025, doi:10.3390/bs15060742_

Round 1
Reviewer 1 Report (New Reviewer)
Comments and Suggestions for Authors
Thank you very much for inviting me to revise the manuscript "Training University Psychology Students to Teach Multiple Skills..."
I am a behavioural consultant and researcher, listing a revision point-to-point.
Since Behavioral Skill Training is the core of the study, the authors should include it in the title.
abstract
The authors should include skill domains learned by children in the abstract.
Line 26 is not clear.
31
The authors should update the keywords.
38
repetition ASD extension
The citations do not seem to follow editorial requirements throughout the manuscript.
Line 44
furnish a link, please.
51-53
extend, please.
59-63
Specify that Stimulus Preference Assessment procedures preceded the selection of reinforcement.
Specify the masterization criterion to succeed in the target.
72-80
Other motifs explain the content described, to extend, please.
82
BST extension
88-92
In which skill domains does BST receive evidence? Extend, please.
125-132
Not clear, describe, please.
136-146
the research questions need an improvement in style.
2.1 (paragraphs)
The authors do not provide sufficient information concerning sampling, participant and enrollment strategies, and the ABA curriculum of children.
179-191
I suggest using a Table.
210-212
it is which skill dimensions, targets, and related dosage authors refer to.
223-231
I suggest adding the total agreement and the motif because such parameters are less measured than others.
258-275
Was the trainer a BCBA? What competence does he have?
The authors declare a passage from standard ABA and progressive one. What was your task list and references?
I detected diverse repetitions, and the use of symbols needs revision.
345-351
it is not clear.
PROCEDURE
I suggest adding a flowchart of your research design.
I note confusion about different factors: CONFEDERATE/CHILDREN/SESSIONS/TIME/DOSAGE/DATA/MANIPULATING VARIABLES.
Moreover, since the study is on BST, I suggest adding the integrity data measurement (also in the appendix) and related data recording.
Likewise, how did authors have test bias regarding the achievements of the children? In effect, the incorrect responses were low at the beginning of training. I suppose children practised merely the maintenance.
Why did the author count the total of correct responses?
Table 1
The table needs a revision since the information reported is repetitive.
Have you performed a data analysis?
Figure 3
Why did the author not show the data per domain?
Table 2
The table does not add information since there is a constant.
Discussion
I found the discussion as the section more challenging.
546-558
A bias was the enrollment of participants (dependent subgroup) and children who learned the ABA therapy.
568-590
it is unclear if the authors intend to refer to a guideline and how SCDs are displayed scientifically (Consider supplemental material to furnish websites or lists).
591-605
I found a bit off-topic this section.
635-639
I found an inappropriate conclusion based on your evidence.
668-670
Why missing data not were treated?
730-737
Not clear.
773-783
Not clear.
796-813
What is your approach to respecting innovations?
814-832
this section is a bit redundant.
In my opinion, the discussion needs a reorganization removing redundant content (in different sections seems an introduction)
Concluding,
I appreciate your investigation into adopting BST to train students.
The methodology (non-concurrent MBD, integrity, follow-ups) follows the standard. The results can be improved and the discussion needs a major revision.
I suggest after these changes your manuscript merits consideration for publication.
Good Luck
Author Response
Please see the attachment.

Reviewer 2 Report (New Reviewer)
Comments and Suggestions for Authors
I think thisis a very well researchered and written manuscript. I have concerns about the relevance of this paper for the wider world. ABA has fallen out of favour in Europe and counties such as Australia and others. I would also ask that you think about the discourse around identity forst and person first language, as per comment in the manuscript. However, neither of these comments diminish the quality of the research or writing.

Author Response
Reviewer 2
Dear reviewer, thank you very much for the contributions. We have provided information highlighted in red color in the manuscript.
I think this is a very well researched and written manuscript. I have concerns about the relevance of this paper for the wider world. ABA has fallen out of favor in Europe and counties such as Australia and others. I would also ask that you think about the discourse around identity first and person first language, as per comment in the manuscript. However, neither of these comments diminish the quality of the research or writing.
Reviewer 3 Report (New Reviewer)
Comments and Suggestions for Authors
The article is interesting. The authors have accurately described the general approach to Applied Behavior Analysis interventions for children with autism spectrum disorder (ASD).
The authors indicate that: "Brazil, the number of well-trained professionals is still not sufficient for the growing number of children with ASD who need intensive behavioral interventions (Ávila & Matos, 2023; Matos, Nascimento, et al., 2021). An alternative to this could be training parents and other caregivers to implement interventions to their children themselves, which is also EBP (Schmidt et al., 2024). Leaf et al. (2017b) states that parent training increases the intensity of interventions and may promote improvements in the relationship with their children with ASD. However, involving caregivers in the process of implementing interventions through DTT (e.g., Ávila & Matos, 2023; Fingerhut & Moeyaert, 2022; Higgins et al., 2023) and other more naturalistic formats (e.g., Bagaiolo et al., 2017; 2022; Ferguson et al., 2023; Sena et al., 2024; Tsami et al., 2019) may not be an easy task in Brazil. Many families have a low socioeconomic status, the members spend many hours at work or on household chores and are many times unavailable to be trained (Ávila & Matos, 2023; Gomes et al., 2022). Training programs for a greater number of professionals are warranted to expand the possibilities of access to quality and lower-cost behavioral interventions for children with ASD, whose family members lack the time to be properly trained.”
The authors should review the situation in different countries. I would recommend that they also include a review of European countries. This way, the article would be more consolidated and allow greater access to information from other platforms that cover the same topic.
It might also be interesting to include a line indicating the potential stress for family members when they cannot...
The authors indicate that: "Many families have a low socioeconomic status, the members spend many hours at work or on household chores and are often unavailable for training (Ávila & Matos, 2023; Gomes et al., 2022)." It might also be interesting to include a line indicating the potential stress for family members in this situation.
The research design of the article fulfills the purpose of evaluating the effects of BST on the precise teaching of multiple repertoires through DTT by six psychology university students to a peer and six children diagnosed with ASD. Generalization and maintenance assessments were conducted.
Although the research sample is small, the study allows for statistical inferences to be made across a given population, respecting the statistical requirements for extrapolation.
The analyses of the results are simple but clear and coherent. The data analyses allow for extracting appropriate and effective information from the data set managed by the researchers.
In the "5. Conclusions" section, the authors should expand on the proposal for participation in online training sessions based on remote meetings.
Author Response
Reviewer 3
Dear reviewer, thank you very much for the contributions. We have information highlighted in red color.
The article is interesting. The authors have accurately described the general approach to Applied Behavior Analysis interventions for children with autism spectrum disorder (ASD).
The authors indicate that: "Brazil, the number of well-trained professionals is still not sufficient for the growing number of children with ASD who need intensive behavioral interventions (Ávila & Matos, 2023; Matos, Nascimento, et al., 2021). An alternative to this could be training parents and other caregivers to implement interventions to their children themselves, which is also EBP (Schmidt et al., 2024). Leaf et al. (2017b) states that parent training increases the intensity of interventions and may promote improvements in the relationship with their children with ASD. However, involving caregivers in the process of implementing interventions through DTT (e.g., Ávila & Matos, 2023; Fingerhut & Moeyaert, 2022; Higgins et al., 2023) and other more naturalistic formats (e.g., Bagaiolo et al., 2017; 2022; Ferguson et al., 2023; Sena et al., 2024; Tsami et al., 2019) may not be an easy task in Brazil. Many families have a low socioeconomic status, the members spend many hours at work or on household chores and are many times unavailable to be trained (Ávila & Matos, 2023; Gomes et al., 2022). Training programs for a greater number of professionals are warranted to expand the possibilities of access to quality and lower-cost behavioral interventions for children with ASD, whose family members lack the time to be properly trained.”
The authors should review the situation in different countries. I would recommend that they also include a review of European countries. This way, the article would be more consolidated and allow greater access to information from other platforms that cover the same topic.
- ** We appreciate the recommendation, and we believe it is very important. However, we sought to address gaps in literature pointed out by a recent meta-analysis study conducted by Fingerhut and Moeyaert (2022), which we believe is enough for our purposes.
It might also be interesting to include a line indicating the potential stress for family members when they cannot...
The authors indicate that: "Many families have a low socioeconomic status, the members spend many hours at work or on household chores and are often unavailable for training (Ávila & Matos, 2023; Gomes et al., 2022)." It might also be interesting to include a line indicating the potential stress for family members in this situation.
- ** We have added the following information in the manuscript: “Furthermore, family members, especially parents of children with ASD, commonly find themselves under high levels of stress in these circumstances, which can make it even more difficult for them to adhere to a training process”.
The research design of the article fulfills the purpose of evaluating the effects of BST on the precise teaching of multiple repertoires through DTT by six psychology university students to a peer and six children diagnosed with ASD. Generalization and maintenance assessments were conducted.
Although the research sample is small, the study allows for statistical inferences to be made across a given population, respecting the statistical requirements for extrapolation.
- ** Thank you for the recommendation. We also believe this is relevant, but we also do know that it is not uncommon to have the absence of statistical analysis in studies using single case designs such as ours. This way, we would like not to do it.
The analyses of the results are simple but clear and coherent. The data analyses allow for extracting appropriate and effective information from the data set managed by the researchers.
In the "5. Conclusions" section, the authors should expand on the proposal for participation in online training sessions based on remote meetings.
- ** Dear reviewer, we added the following information in the end:
“Caregivers and trainers will not need to be physically in the same place. All four BST training components may be administered remotely. The trainers may record didactic instructions in audio and video on multiple repertoires that caregivers will have to teach to their children with ASD. During the modeling component, an experimenter may record videos of two actors representing the roles of interventionist and child with ASD rehearsing DTT. Caregivers will be able to download the videos to watch whenever they need. During the role-play component, caregivers will conduct DTT directly with their children with ASD, and the experimenter, after seeing the process through videoconferencing in his/her smartphone, will provide performance feedback to increase the caregivers’ accuracy in teaching their children. As performance improves, generalization and maintenance assessment conditions in the long term may be planned and established”.

Reviewer 4 Report (New Reviewer)
Comments and Suggestions for Authors
This report presents a review of the article Training University Psychology Students to Teach Multiple Skills to Children with Autism Spectrum Disorder by Daniel Carvalho de Matos et al.
The article is well structured and provides an in-depth report on the assessment of the effects of behavioural skills training on the accurate teaching of multiple repertoires using discrete trial training by six Psychology university students to a confederate and six children diagnosed with ASD. The article contributes to the existing body of knowledge in the area of the use of behavioural skills training in the field of autism research. The authors present a comprehensive approach in this single case design using nonconcurrent multiple baselines across participants but should also identify their own positionality in the research. The study involved multiple points of data collection involving a range of participants from the trainee psychology students to the participating children. The use of interobserver agreement procedures enhanced the validity of the study components and the figures provided clarity in terms of the findings of the DTT procedures. Overall the article is well organised but there are some areas that need attention where clarity, methodological rigour and analysis could be strengthened.
The literature reviewed supported the identification of a research gap however it is also appropriate to discuss a broad range of the literature around the topic under scrutiny. Internationally ABA has come under a spotlight as having negative effects on individuals with ASD. This should be noted in the article and mitigating factors presented. See report below as example: Leaf, J.B., Cihon, J.H., Leaf, R. et al. Concerns About ABA-Based Intervention: An Evaluation and Recommendations. J Autism Dev Disord 52, 2838–2853 (2022). https://doi.org/10.1007/s10803-021-05137-y
The absence of pre-test data on the specific DTT teaching sessions with children with ASD is noted as a limitation by the authors but it places a question over the reliability of the responses given in the data. The authors need to provide specifics about the student responses and how these are specific to the training and whether the students already had these specific skills which presents a methodological inaccuracy.
In terms of ethics, international guidelines recommend that children have a right to have their voice heard in research. In the study authors highlight that informed consent was provided however they do not address informed assent from the child participants, which would enhance the ethical considerations element.
In the discussion section the authors move away from the findings of the study and instead discusses literature on remote BST training. While it is relevant there is an over emphasis given considering this is not the study focus. More clarity is needed in the discussion section to keep the focus on the current study.
Author Response
Reviewer 4
Dear reviewer, thank you very much for the contributions. We have information highlighted in red color.
This report presents a review of the article Training University Psychology Students to Teach Multiple Skills to Children with Autism Spectrum Disorder by Daniel Carvalho de Matos et al.
The article is well structured and provides an in-depth report on the assessment of the effects of behavioural skills training on the accurate teaching of multiple repertoires using discrete trial training by six Psychology university students to a confederate and six children diagnosed with ASD. The article contributes to the existing body of knowledge in the area of the use of behavioural skills training in the field of autism research. The authors present a comprehensive approach in this single case design using nonconcurrent multiple baselines across participants but should also identify their own positionality in the research. The study involved multiple points of data collection involving a range of participants from the trainee psychology students to the participating children. The use of interobserver agreement procedures enhanced the validity of the study components and the figures provided clarity in terms of the findings of the DTT procedures. Overall the article is well organised but there are some areas that need attention where clarity, methodological rigour and analysis could be strengthened.
The literature reviewed supported the identification of a research gap however it is also appropriate to discuss a broad range of the literature around the topic under scrutiny. Internationally ABA has come under a spotlight as having negative effects on individuals with ASD. This should be noted in the article and mitigating factors presented. See report below as example: Leaf, J.B., Cihon, J.H., Leaf, R. et al. Concerns About ABA-Based Intervention: An Evaluation and Recommendations. J Autism Dev Disord 52, 2838–2853 (2022). https://doi.org/10.1007/s10803-021-05137-y
- ** Dear reviewer. We have added the following information at the end of the discussion section.
“Although the current study identified and sought to address gaps identified in previous research on BST training for DTT implementation (Fingerhut & Moeyaert, 2022), ABA applied to ASD has been targeted of criticisms by some communities, that is, some autism rights and neurodiversity activists. According to Leaf, Cihon, Leaf, et al. (2022) there are some concerns that underlie the criticisms such as, for example,: Concerns about Ivar Lovaas and his UCLA Young Autism Project, which improved the quality of life of those with ASD, but, in the past, it used moderate punishment strategies for a brief period to decrease interfering behaviors. Practices based on the use of positive reinforcement have always, however, been defined as a priority. Another concern refers to the intensity of interventions. Some people may advocate, for example, that 40 hours of intervention per week is an exaggeration. However, Leaf, Cihon, Leaf, et al. advocates that the number of hours per week must be based upon individual needs, and that there is no research to support that any intervention intensity may be harmful for learners.
Considering the research from this manuscript, the participating children in the university laboratory, where data collection took place, only attended the laboratory once a week for 1 hour and 30 minutes, regardless of the study. This happened so the University could provide ABA services for a higher number of children, whose parents and caregivers are unable to pay high prices for these services. Even though, most of the children developed several important repertoires, which helped them to be more functional and independent. It has also been suggested by some research that ABA may lead to negative outcomes such as depression, post-traumatic stress disorder (PTSD) and anxiety. However, Leaf, Cihon, Leaf, et al. (2022) discussed that these studies are based on reports, which lack in reliability or replication. Despite criticism by some groups of individuals, ABA is characterized for many years by producing scientific evidence of the effectiveness of its practices, which also includes, for example, BST and DTT. It is also discussed that continued examination of the applications to solve human significant issues is warranted to refine the applications or replace them for new and better ones”.
The absence of pre-test data on the specific DTT teaching sessions with children with ASD is noted as a limitation by the authors but it places a question over the reliability of the responses given in the data. The authors need to provide specifics about the student responses and how these are specific to the training and whether the students already had these specific skills which presents a methodological inaccuracy.
- ** We acknowledge the limitations. We have provided new pictures providing information on the students’ skills and responses. We understand the problem of methodological inaccuracy and we discussed it in the new version of the manuscript.
In terms of ethics, international guidelines recommend that children have a right to have their voice heard in research. In the study authors highlight that informed consent was provided however they do not address informed assent from the child participants, which would enhance the ethical considerations element.
- ** We have added the following information:
The children themselves signed an informed assent to participate. Everything was in accordance with Resolution 510, of April 7, 2016, of the National Health Council in Brazil.
In the discussion section the authors move away from the findings of the study and instead discusses literature on remote BST training. While it is relevant there is an over emphasis given considering this is not the study focus. More clarity is needed in the discussion section to keep the focus on the current study.
We changed some parts of the discussion, but, when we were discussing remote BST, we were concerned with the possibility of new systematic replication studies. We do not believe we moved away from the theme.

Reviewer 5 Report (New Reviewer)
Comments and Suggestions for Authors
This is a very interesting and well written article and I thank you for the opportunity to read it. It makes a valuable contribution to the intersection of psychology and autism intervention by exploring how psychology students can be effectively trained in BST to support skill development in children with ASD through DTT. It addresses an important and timely gap in professional preparation, offering potential pathways for improving inclusive practices and workforce readiness in the field of psychology. This is all aligned to the literature that suggests that due to the lack of access to professionals in the area, caregivers and paraprofessionals can engage in supports such as DTT. I particularly commend you on the rigorous maintenance and generalisation approaches carried out over the four month period.
I have very few observations for you to consider:
- Give a brief explanation of ABA in the introduction, just to contextualise the your argument.
- Put the baseline section - lines 282 -302 into a table format for ease of reading.
- Line 844 - explain what social validity is in the context of the BST -i.e. that it is considered valuable and acceptable by all involved.
- A little more information on ethical procedures warranted - data stored in a secure place? adherence to GDPR procedures? who had access to the data? all information de-identified. use of pseudonyms. I think these are extremely important particularly when working with vulnerable children.
- Why was a period of four months chosen? Give a brief rationale for this timeframe.
- Was there any opportunity for the psychology students to give their perspective on the BST and implementation of the DTT? Would it be good to add this or note this as something for further study / a limitation of the current one?
- Likewise, was there any qualitative feedback or survey responses from the children with ASD on their experience? It is always good to get the student voice, particularly students with disability. Again, this could be noted for further research / limitation of current study.
- Line 703 -"People in need to learn how to implement proper ABA interventions may en- " - is the word 'in' surplus here?
- Proof read for very minor errors such as point 8.
Author Response
Reviewer 5
Dear reviewer, thank you very much for the contributions. We have information highlighted in red color.
Comments and Suggestions for Authors
This is a very interesting and well written article and I thank you for the opportunity to read it. It makes a valuable contribution to the intersection of psychology and autism intervention by exploring how psychology students can be effectively trained in BST to support skill development in children with ASD through DTT. It addresses an important and timely gap in professional preparation, offering potential pathways for improving inclusive practices and workforce readiness in the field of psychology. This is all aligned to the literature that suggests that due to the lack of access to professionals in the area, caregivers and paraprofessionals can engage in supports such as DTT. I particularly commend you on the rigorous maintenance and generalisation approaches carried out over the four month period.
I have very few observations for you to consider:
Give a brief explanation of ABA in the introduction, just to contextualise the your argument.
- ** ABA represents one of the three branches of the science of behavior analysis (Behaviorism; Experimental Analysis of Behavior; Applied Behavior Analysis). ABA focuses on changing socially significant behavior (Leaf, Cihon, Leaf, et al., 2022).
Put the baseline section - lines 282 -302 into a table format for ease of reading.
- We have added a figure representing a flowchart of the research conditions.
Line 844 - explain what social validity is in the context of the BST -i.e. that it is considered valuable and acceptable by all involved.
- According to De Souza et al. (2025), social validity allows one to know if interventions are based on meaningful goals, acceptable strategies and involve positive outcomes to those who are target of the interventions and stakeholders.
A little more information on ethical procedures warranted - data stored in a secure place? adherence to GDPR procedures? who had access to the data? all information de-identified. use of pseudonyms. I think these are extremely important particularly when working with vulnerable children.
** This research was approved by the human research ethics committee of the Federal University of Maranhão (authorization 4.284.271). The six university students, the six children with ASD and those responsible for them signed an informed consent form. The children themselves signed an assent consent to participate. Everything was in accordance with Resolution 510, of April 7, 2016, of the National Health Council in Brazil. All personal information was confidential. The participants (university students and children) could remove their consent or assent at any time, if they wished, without any harm. All participants have the right to know the rationale, objectives and procedures to be used, with information on methods provided in clear and accessible language. Precautions to prevent any harm should be made explicit. Participants are guaranteed confidentiality and privacy throughout the research process. The benefits of the research should be made explicit. Participants are guaranteed access to the research results whenever they wish. They are also guaranteed reimbursement and forms of coverage for expenses arising from the research, if any. Participants receive information about the ethics committee for research involving human beings, including telephone contact, which processed the research protocol. Participants also have the right to access the consent and assent record whenever they requested them.
Why was a period of four months chosen? Give a brief rationale for this timeframe.
- ** It is important to say that the period of the maintenance condition was four months because the Psychology students, after BST training in the University laboratory where data collection took place, collaborated as interns for at least a semester. After that, they could still collaborate, if they wished, as volunteers. It is also important to say that their training, anyway, was not compulsory. They could remove their consent to participate at any time without any harm, if they wished.
Was there any opportunity for the psychology students to give their perspective on the BST and implementation of the DTT? Would it be good to add this or note this as something for further study / a limitation of the current one?
Likewise, was there any qualitative feedback or survey responses from the children with ASD on their experience? It is always good to get the student voice, particularly students with disability. Again, this could be noted for further research / limitation of current study.
** Still regarding social validity, it was said that the university students rated their training solely through a questionnaire with objective questions in the Likert format. However, it is important that future studies also allow university participants to give their perspectives on the BST and implementation of DTT with comments for a more qualitative interpretation. Likewise, future studies may also obtain qualitative feedback from participating children with ASD, which also did not happen in the current research.
Line 703 -"People in need to learn how to implement proper ABA interventions may en- " - is the word 'in' surplus here?
- ** People interested in learning how to implement proper ABA interventions
Proof read for very minor errors such as point 8.

Round 2
Reviewer 1 Report (New Reviewer)
Comments and Suggestions for Authors
Thank you for inviting me to revise the current manuscript.
The authors have followed the reviewers' suggestions to improve the manuscript's clarity and quality/method.
After having read the revised version, I will suggest minor changes.
Abstract
I am concerned about whether it is better to specify that the children's educational curriculum was maintained (see line 201).
Introduction
Check the citations, please.
70-72
During the provision of DTT, the learner is exposed to different targets as he or she meets arbitrary learning criteria (Leaf, Cihon, Ferguson, et al., 2022; Souza & Ribeiro, 2023; Varella & Souza, 2018).
I suggest introducing the undesirable effects of such interventions, such as caregiver and client stress, and how BACB provides ethical codes for compassionate and tailored behavioral interventions.
Line 98
Remove the “behavioral skill training”, leaving only BST if previously cited.
The authors have added benefits of applying BST in several domains and for different practitioners, and rewritten unclear sections.
Materials and Methods
The authors have provided information about the recruitment of participants.
Line 186-195
Consider (P1-P6) and (CD1-CD6)
253
Check the meaning, please.
270
Consider removing (P1, P2, P3, P4, P5 and P6) or leaving (P1-P6).
275-281
Ensure that such a difference in IOA rating is discussed further.
286-287
Consider adding recent citations about SCDs, since there is a debate on their methodological value.
313
At follow-up, will the child remain the same for each confederate?
Figure 1
A six-column chart with arrows would be better than a vertical one in such a case.
422-423
The experimenter administered four BST components (didactic instruction, modeling, role-play, and performance feedback) to train each of the six university participants.
Consider removing such a statement, as it makes the manuscript redundant.
425
sitting still, motor imitation, making requests, vocal imitation, and receptive identification of non-verbal stimuli) were given.
Idem
Consider reporting only “BST” without the four components after the first description throughout the manuscript, increasing the readability.
461-463
As was said, five of these repertoires (sitting still, motor imitation, making requests, vocal imitation, and receptive identification of non-verbal stimuli) were involved in the training conditions through BST.
Idem
497
Idem
560
Please check if the note should appear under the table.
Table 2
Rows: BL or LB? (baseline), check please.
A dispersion graph could be more appropriate since the data are discrete/percentage and in series (DECREASING).
Likewise, the authors can replace the data with percentages of improvements (INCREASING).
I also had doubts about this table in the previous revision.
In my opinion, it merits an improvement.
Did the author plan correlation analyses?
I detect such an opportunity.
561
Idem
I thank the authors for having furnished the data about the maintenance of children in x domains.
The total number of correct responses (CD1, CD2, CD3…, etc.) for maintenance does not show variability.
In effect, the result of Figure 4 is a bit informative.
Consider improving data visualization in general.
Theoretically, “As seen in Figure 4, CD1 accumulated 2047 correct and only incorrect responses…
The children would have shown the same result in other settings/people since mastering skills.
Conversely, the core of your investigation is the relationship between the rise and fall of students' procedural integrity (DTT and BST) and children's outcomes.
I feel you could develop such a research hypothesis as the core of your investigation.
654-657
According to Figure 5, CD1 emitted 1-a total of 155 correct responses and one incorrect response in the sitting still program; 2-a total of 156 correct responses and one incorrect response in the eye contact program; 3-a total of 169 correct responses and one incorrect response in the retelling of stories program; 4-a total of 160 correct responses and…
Why did the author report the result as a total rather than a percentage of correct responses?
For example, out of the total (10 trials), 90% of the children performed (for all sessions).
Accumulative scores (merge of several domains) lack qualitative/functional data collection information.
Consider trying to align graphs by domain (I note that such a child had more competencies)
SITTING STILL (1) – (2) – (3)
(4) – (5) – (6)
The extra domains not covered by all children could be described merely in the text, providing major clarity to the manuscript.
Consider adding the data from students to the graphs (signaling them with the second axis)
Your result section merits an improvement, aligning most of the research hypotheses.
Have you considered expanding the social validity interview with retrospective questions to students? (to improve clinical information)
Enjoyed participating in the BST training (Which domain was more challenging to follow?)
I felt comfortable with the training process (how many have you thought of continuing the training in ABA? What are the pros and cons?).
I learned essential skills (were there some skills where the student felt anxious?.
BST effectively taught children with ASD accurately (What idea did students formulate about ABA?).
I will continue to use the procedures I mastered to teach other skills to children with ASD (did the student think other clinical samples could need similar procedures?).
I recommend the training to other interested people (in what manner, similar procedure could improve my living skills or quality of life? (also regarding relatives and partner)
Discussion
820
Overall, the children accumulated more than 1000 correct responses across several sessions.
Idem
831
But also to measure the potential of BST to influence significant developmental gains for children with ASD.
The results that I intend to improve.
849
who also did not exhibit significant levels of disruptive/interfering behaviors.
The authors also explained this previously. However, another point was that the same DTT could lose sense with severely challenging behaviors. In such cases, the students would have received training on other EBPs. Concluding that the bias regarding the sample was a convenience sample.
Consider referring to your hypothesis to show major rigor in the discussion before starting the study's limitations.
1) Did BST effectively train university students to accurately teach multiple skills via DTT to a confederate pretending to be a child with ASD?
2) Did BST training produce the generalization of accurate teaching of multiple repertoires, including new ones not covered in training, to children with ASD across post-training probes?
3) After BST training and assessment of generalization proved successful, would the university students' high teaching integrity levels during the provision of DTT and the children's acquisition of repertoire be demonstrated across four months on average?
The discussion includes several topics about telehealth, video modeling, unserved areas, DTT limitations, SCDs, and surrounding ABA implementations. However, the authors could reduce redundancies and dedicate efforts to discussing the BST package, its modules, and development.
- In which clinical or non-clinical area could future applications include such a procedure?
- Which module can be implemented independently?
- For what learning areas are single modules and their development suitable?
- How can Component Analysis study the effect of BST or a single module?
I hope my suggestion will increase the quality and dissemination of your study.
Good Luck
Some inputs.
Hassan M, Simpson A, Danaher K, Haesen J, Makela T, Thomson K. An Evaluation of Behavioral Skills Training for Teaching Caregivers How to Support Social Skill Development in Their Child with Autism Spectrum Disorder. J Autism Dev Disord. 2018 Jun;48(6):1957-1970. doi: 10.1007/s10803-017-3455-z. PMID: 29307038.
Flowers, J., & Cuitareo, J. (2023). Behavioral Skill Training: A Single-Case Meta-Analysis. Journal of Human Services: Training, Research, and Practice, 9(2), 4.
Author Response
Reviewer 1
Dear reviewer, thank you very much for the contributions. We have information highlighted in green color.
Comments and Suggestions for Authors
Thank you for inviting me to revise the current manuscript.
The authors have followed the reviewers' suggestions to improve the manuscript's clarity and quality/method.
After having read the revised version, I will suggest minor changes.
Abstract
I am concerned about whether it is better to specify that the children's educational curriculum was maintained (see line 201).
- ** But it was said that the children were taught skills related to their individualized curriculum goals across four months, after the university students were trained.
Introduction
Check the citations, please.
70-72
During the provision of DTT, the learner is exposed to different targets as he or she meets arbitrary learning criteria (Leaf, Cihon, Ferguson, et al., 2022; Souza & Ribeiro, 2023; Varella & Souza, 2018).
I suggest introducing the undesirable effects of such interventions, such as caregiver and client stress, and how BACB provides ethical codes for compassionate and tailored behavioral interventions.
- ** Done.
Line 98
Remove the “behavioral skill training”, leaving only BST if previously cited.
The authors have added benefits of applying BST in several domains and for different practitioners, and rewritten unclear sections.
- ** Done.
Materials and Methods
The authors have provided information about the recruitment of participants.
Line 186-195
Consider (P1-P6) and (CD1-CD6)
- ** Done.
253
Check the meaning, please.
- ** We do not understand what else to do.
270
Consider removing (P1, P2, P3, P4, P5 and P6) or leaving (P1-P6).
- ** Done.
275-281
Ensure that such a difference in IOA rating is discussed further.
286-287
Consider adding recent citations about SCDs, since there is a debate on their methodological value.
313
At follow-up, will the child remain the same for each confederate?
- ** Yes. We had the following dyads: P1 – CD1; P2 – CD2; P3 – CD3; P4 – CD4; P5 – CD5; P6 – CD6.
Figure 1
A six-column chart with arrows would be better than a vertical one in such a case.
- ** Done.
422-423
The experimenter administered four BST components (didactic instruction, modeling, role-play, and performance feedback) to train each of the six university participants.
Consider removing such a statement, as it makes the manuscript redundant.
- ** Done.
425
sitting still, motor imitation, making requests, vocal imitation, and receptive identification of non-verbal stimuli) were given.
Idem
- ** Done.
Consider reporting only “BST” without the four components after the first description throughout the manuscript, increasing the readability.
461-463
As was said, five of these repertoires (sitting still, motor imitation, making requests, vocal imitation, and receptive identification of non-verbal stimuli) were involved in the training conditions through BST.
Idem
- ** Done.
497
Idem
- ** Done.
560
Please check if the note should appear under the table.
- ** Fixed.
Table 2
Rows: BL or LB? (baseline), check please.
- ** BL
A dispersion graph could be more appropriate since the data are discrete/percentage and in series (DECREASING).
Likewise, the authors can replace the data with percentages of improvements (INCREASING).
I also had doubts about this table in the previous revision.
In my opinion, it merits an improvement.
Did the author plan correlation analyses?
I detect such an opportunity.
561
Idem
- ** We replaced the table with a figure with scatter plots showing the percentage of implementation errors of DTT components by each participant. We just wanted to show that, after BST, errors were significantly reduced. During the last condition (maintenance), they were reduced to zero.
I thank the authors for having furnished the data about the maintenance of children in x domains.
The total number of correct responses (CD1, CD2, CD3…, etc.) for maintenance does not show variability.
In effect, the result of Figure 4 is a bit informative.
Consider improving data visualization in general.
Theoretically, “As seen in Figure 4, CD1 accumulated 2047 correct and only incorrect responses…
The children would have shown the same result in other settings/people since mastering skills.
Conversely, the core of your investigation is the relationship between the rise and fall of students' procedural integrity (DTT and BST) and children's outcomes.
I feel you could develop such a research hypothesis as the core of your investigation.
- ** However, interactions with the children with ASD solely occurred during generalization and maintenance conditions. In Baseline and BST conditions, the university students only taught a confederate pretending to act like a child with ASD. High teaching integrity levels by the students generalized and maintained across four months when they taught the children. The students were able to evoke correct responses by the children in both generalization and maintenance conditions. Perhaps interactions for teaching children should have happened since baseline condition, but we preferred to avoid that the students, still with little experience in the beginning, tried to teach children with ASD.
654-657
According to Figure 5, CD1 emitted 1-a total of 155 correct responses and one incorrect response in the sitting still program; 2-a total of 156 correct responses and one incorrect response in the eye contact program; 3-a total of 169 correct responses and one incorrect response in the retelling of stories program; 4-a total of 160 correct responses and…
Why did the author report the result as a total rather than a percentage of correct responses?
- ** We have added percentages too.
For example, out of the total (10 trials), 90% of the children performed (for all sessions).
Accumulative scores (merge of several domains) lack qualitative/functional data collection information.
Consider trying to align graphs by domain (I note that such a child had more competencies)
SITTING STILL (1) – (2) – (3)
(4) – (5) – (6)
The extra domains not covered by all children could be described merely in the text, providing major clarity to the manuscript.
- ** We have kept the figures by domains the way they were because only three domains were covered by all children (sitting still, making eye contact and answering questions). It was not most of them.
Consider adding the data from students to the graphs (signaling them with the second axis)
Your result section merits an improvement, aligning most of the research hypotheses.
Have you considered expanding the social validity interview with retrospective questions to students? (to improve clinical information)
Enjoyed participating in the BST training (Which domain was more challenging to follow?)
I felt comfortable with the training process (how many have you thought of continuing the training in ABA? What are the pros and cons?).
I learned essential skills (were there some skills where the student felt anxious?.
BST effectively taught children with ASD accurately (What idea did students formulate about ABA?).
I will continue to use the procedures I mastered to teach other skills to children with ASD (did the student think other clinical samples could need similar procedures?).
I recommend the training to other interested people (in what manner, similar procedure could improve my living skills or quality of life? (also regarding relatives and partner)
- ** We believe that these suggestions for the social validity interview are relevant for research in the future.
Discussion
820
Overall, the children accumulated more than 1000 correct responses across several sessions.
Idem
831
But also to measure the potential of BST to influence significant developmental gains for children with ASD.
The results that I intend to improve.
849
who also did not exhibit significant levels of disruptive/interfering behaviors.
The authors also explained this previously. However, another point was that the same DTT could lose sense with severely challenging behaviors. In such cases, the students would have received training on other EBPs. Concluding that the bias regarding the sample was a convenience sample.
Consider referring to your hypothesis to show major rigor in the discussion before starting the study's limitations.
1) Did BST effectively train university students to accurately teach multiple skills via DTT to a confederate pretending to be a child with ASD?
2) Did BST training produce the generalization of accurate teaching of multiple repertoires, including new ones not covered in training, to children with ASD across post-training probes?
3) After BST training and assessment of generalization proved successful, would the university students' high teaching integrity levels during the provision of DTT and the children's acquisition of repertoire be demonstrated across four months on average?
The discussion includes several topics about telehealth, video modeling, unserved areas, DTT limitations, SCDs, and surrounding ABA implementations. However, the authors could reduce redundancies and dedicate efforts to discussing the BST package, its modules, and development.
- In which clinical or non-clinical area could future applications include such a procedure?
- Which module can be implemented independently?
- For what learning areas are single modules and their development suitable?
- How can Component Analysis study the effect of BST or a single module?
- ** We hope to systematically replicate our study in the future with parents and other caregivers using telehealth through videoconferencing from cell phones with internet. We plan to assess if the remote BST will successfully train the caregivers to teach multiple repertoires, via DTT with high performance accuracy, to their children. We also plan to assess if performance accuracy will generalize during the teaching of new skills. We will assess maintenance to check if the BST effects will be long lasting and we will systematically monitor gains in repertoire by the children. We want this possible new investigation to be like what we did in the research with the university students, but remotely and involving caregivers as participants. We do not wish to conduct a separate component analysis since it has been discussed that training is more effective when the components are used together.
I hope my suggestion will increase the quality and dissemination of your study.
Good Luck
Some inputs.
Hassan M, Simpson A, Danaher K, Haesen J, Makela T, Thomson K. An Evaluation of Behavioral Skills Training for Teaching Caregivers How to Support Social Skill Development in Their Child with Autism Spectrum Disorder. J Autism Dev Disord. 2018 Jun;48(6):1957-1970. doi: 10.1007/s10803-017-3455-z. PMID: 29307038.
Flowers, J., & Cuitareo, J. (2023). Behavioral Skill Training: A Single-Case Meta-Analysis. Journal of Human Services: Training, Research, and Practice, 9(2), 4.
This manuscript is a resubmission of an earlier submission. The following is a list of the peer review reports and author responses from that submission.
Round 1
Reviewer 1 Report
Comments and Suggestions for Authors
The study provided an important insight into training psychology students to teach children with autism using behavioral skills training (BST), and showed positive results in terms of students’ ability to teach multiple skills with high accuracy, with this effect lasting even after several months. Students also rated the training highly, reflecting their satisfaction with the experience. However, despite these encouraging results, there are some points that require consideration. First, the study focused on a limited number of students and children, which raises questions about the extent to which the results can be generalized to a wider scale. Second, the actual challenges that students may face when applying these skills in more complex environments, such as working with children with more diverse needs and behaviors, were not sufficiently addressed. Also, it is good to know that the children showed improvement and recorded more than 1,000 correct responses, but the question here is: Was this improvement sustainable in the long term? And was it reflected in their daily lives in a practical way? The study would have been more valuable if it included a follow-up with the children after a longer period to see how the training continued to affect their life skills development. Overall, the study adds clear scientific value and confirms the effectiveness of BST in preparing students to teach children with autism, but it also opens the door to further research into how this training can be improved to be more comprehensive and applicable in more realistic settings.
The introduction section was highly sufficient for enriching the story behind ABA use in training ASD children to acquire an adequate useful behavior, while presenting BST (applied with psychology students) as a new concept inspired from DTT (applied with autistic children by the previously trained students).
Well illustrated and clear results section.
The results confirm the effectiveness of behavioral skills training (BST) in improving psychology students’ ability to teach children with autism, reflecting a long-term positive impact on their daily skills. The results are an important contribution to enhancing the quality of behavioral interventions, especially with the potential for generalization of the training to include a larger number of interventionists and reduce costs for families.
However, it is recommended to take into account the limitations of the study, such as the small number of weekly sessions and the lack of use of standardized developmental assessment tools, in addition to the need to explore more comprehensive training alternatives, such as the use of smartphones to expand the scope of benefit from BST.
Comments on the Quality of English LanguageSo based on the above, I declare this study to be acceptable for its important scientific contribution to the field of training behavioral interventionists and teaching children with autism.
Author Response
Dear reviewer,
Thank you very much for the important considerations and recommendations. All changes and additions in the new version of the manuscript are highlighted in red color. We also added several comments in this cover letter after your review.
- We understand that the number of participants was limited. However, since our research involved a single-case design, we know that this type of design commonly involves a small number of participants and each of them serves as own control. Generality of the findings may be demonstrated in future systematic replication studies from different research laboratories.
- In fact, previous studies, including ours, show limitations regarding the definition of certain learners diagnosed with ASD to whom DTT may be implemented. According to recent meta-analysis (Fingerhut & Moeyaert, 2022) and systematic review study (De Souza et al., 2025), minimally vocal learners with ASD, who show significant delays in repertoire, have not been targets of research involving BST and DTT, or other investigations involving single-case designs. Regarding the changes and additions to the new version of the manuscript, we have included a discussion on the need for more research with this population as a way of investigating the generality of BST to train multiple skills through DTT. We believe that future studies may conduct a systematic replication of our investigation, extending it by systematically measuring multiple repertoire gains in minimally vocal children with ASD.
- In our study, children were only taught multiple repertoires in a room from a university laboratory on ASD, in which DTT is systematically applied to address skill deficits. Regarding the additions to our manuscript after review, we have discussed that a distraction-free environment, such as the one from our research, is important for learners who demand a comprehensive individualized curriculum to teach multiple skills. However, we recommend that future studies also assess the generality of training (BST) and the teaching of multiple skills in more naturalistic environments, such as residence, which may more accurately reflect the learners’ day-to-day lives. This may be especially the case of learners who show more accurate attending behaviors and more complex language skills, which tend to be pre-requisites to the use of more wordy and complex instructions to teach them, and to the use of varied topographies of instructions and less intrusive prompt types and prompting systems.
- In our study, we formally assessed social validity concerning the impact of BST training for the participating university students. All of them rated BST highly. However, as a limitation, we discussed that social validity measures by the children’s parents or other caregivers, regarding skill development by the children, were not consistently obtained. Nevertheless, the parents and other caregivers of the participating children with ASD in this investigation informally said that the children showed improvements that reflected their functioning in their day-to-day environments. More information concerning this was added to our manuscript after review. We believe that the suggestion of a follow up with the children after a long period is very important, but, after the university students were trained, the children were taught by them with no interruption across four months. Up to the current date, the children are still taught in our laboratory (but no longer by the same university students from the research).
- We added new information to the discussion section regarding limitations and recommendations for new research, after careful review. Please, identify them. We have highlighted all changes and additions in red color.

Reviewer 2 Report
Comments and Suggestions for Authors
Thank you for the opportunity to review: Training University Psychology Students to Teach Multiple 3 Skills to Children with Autism Spectrum Disorder. The authors conducted a multiple baseline design, single case study investigating the use of BST to teach students DTT procedures with autistic children. The authors organization of the manuscript is well done and the writing and tone is consistent with APA scholarly voice.
There are a few wording corrections I would suggest, such as line 62 where it states parent-mediated interventions are "considered" evidence based practice, PMIs ARE evidence based practices, the word "consider" is not direct.
There are several instances of using outdated references (e.g., Baer et al., 1968, Lovaas 1987, Cooper et al., 2014, Skinner, 1992). There are more recent and applicable studies that would substantiate the authors statements better.
While I applaud these authors work, I do not believe this work is adding something new to the literature base. Behavior skills training (BST) is already well researched as an effective model of teaching. The outcome of "Each child accumulated over 1000 correct responses across several sessions" is overstated. This is a very proximal outcome and there is lack of control over other factors that could have led to this result. Additionally, does 1000 correct responses equate to meaningful outcome? A more distal outcome would be more telling here.
Additionally the use of single case design significantly limits the generalizability of this work. A group design study that utilizes distal outcomes would bring more novelty to this study.
Author Response
Dear reviewer,
Thank you very much for the important considerations and recommendations. All changes and additions in the new version of the manuscript are highlighted in red color. We also added several comments in this cover letter after your review.
- We followed the suggestion on wording corrections regarding parent-mediated interventions as evidence-based practices.
- We added recent references on evidence-based practices in the field of Behavior Analysis for ASD and other areas of research (Creem et al., 2022; Leaf, Cihon, et al., 2022; Leaf, Ferguson, et al., 2022; Steinbrenner et al., 2020). Important comments related to them were added through the introduction and discussion sections of the new version of the manuscript. We also kept some old references. Baer et al. (1968) presents the dimensions of ABA, which are important for the training of applied behavior analysts to solve human issues. We replaced the second edition of Cooper et al. (2014) with the third edition (Cooper et al., 2020), which presents relevant content on single case designs, including the case used in our study.
- Although the previous literature already established BST as evidence-based practice, there are still limitations that justify carrying out new studies. In a recent meta-analysis by Fingerhut and Moeyaert (2022), the authors identified 46 studies that used single case designs to assess the effectiveness of BST (or some of its components) to train different individuals (parents, caregivers, professionals and paraprofessionals) to implement DTT to children diagnosed with autism spectrum disorder (ASD) accurately. It was discussed that, when the four components of BST are implemented together, they are statistically significantly effective and that, on average, the participants from the studies in the meta-analysis demonstrated teaching integrity of 96.06% DTT components implemented correctly. On the other hand, it is important to mention that the studies did not include consistent information on data from the learners with ASD. In this sense, it is reasonable to assume that more research on BST is warranted especially to assess its effects on the systematic repertoire acquisition by learners with ASD. In our study, we sought to do this. We sought to fill a gap in literature.
- Previous studies on BST and DTT, involving single case designs, also did not assess accurate teaching of multiple skills, that is, at least ten to children or confederates. Gomes et al. (2022), who conducted a longitudinal study and did not use a single case design, trained and supervised parents and other caregivers of children with ASD. The caregivers conducted intensive behavioral interventions to teach repertoire to their children across a year. It was noticed that children, who were taught at least ten skills systematically, showed more significant improvements in development, as confirmed by standardized development assessment protocols. In our study, we used a single case design to assess the effects of BST on the accurate teaching of multiple skills through DTT. After training and generalization conditions were finished, six university students systematically taught at least ten different skills to six children with ASD, according to their individualized curricular goals, across four months. We did not identify another study, involving single case design, that did this specifically.
- Recent systematic review by De Souza et al. (2025) identified 59 research articles published in Brazilian and international journals from 2007 to 2024, and that investigated the effectiveness of single case methodology on the modification of socially important behaviors of individuals with ASD. De Souza et al. (2025) also assessed the quality of the studies based on the indicators of the “What Works Clearinghouse” (WWC, 2022). It was noticed that, out of 59 articles, only 14 fully followed the WWC standards.
- Our study was conducted in Brazil. We followed the standards for single-subject studies by WWC (2022), seeking to address a problem indicated in the systematic review by De Souza et al. (2025), in the sense that many Brazilian studies from the review did not follow the WWC standards. In other words, in the current investigation, data were presented in graph and table format; the independent variable (IV – BST) was systematically manipulated; interobserver agreement (IOA) was obtained by two observers across all experimental conditions and was determined above 20% of the sessions per condition; the IOA level between the observers was determined above 80% across the experimental conditions; it was used a three-tier multiple baseline for each of the two triads of university students who participated; each of the study’s phases involved at least three data points with little or no variability; also, the research involved the assessment of generalization, maintenance and social validity of procedures and results. Everything was done with the aim of ensuring the internal and external validity of procedures and results, which is expected from single case studies.
- We do not agree with the claim that the cumulative number of correct responses of children with ASD in our study is overstated. Figure 3 allows a comparison between the number of cumulative correct responses versus the number of cumulative incorrect responses by the children with ASD. These data were produced from the teaching carried out by university students, who learned how to perform DTT with the children with high integrity. Before the onset of the study, the university students had no experience with DTT and ASD. Probe sessions, across the four months during which they conducted DTT with the children (after BST training), showed that the university students demonstrated no teaching integrity errors. As said before, previous studies on BST to implement DTT to children with ASD, identified in a meta-analysis research report by Fingerhut and Moeyaert (2022), did not include consistent information on data from the learners with ASD. However, we included consistent information on data from the learners with ASD in our study. Therefore, we argue that the gains in repertoire demonstrated by the children in our study do represent significant data, and these gains should be interpreted as a measure of the effectiveness of the BST, in addition to the improvement in the level of integrity with which university students implemented DTT to the children.
- Single case designs are commonly used in research on Applied Behavior Analysis (ABA) and represent an important path to determine the effectiveness of interventions in producing socially relevant behavior change. We sought to follow parameters established by WWC (2022) for single case studies. Plus, we assessed generalization of high teaching integrity by the university students while conducting DTT with the children with ASD involved. By doing this, we were following one of the important parameters by WWC (assessment of generalization). Besides, future systematic replication investigations may assess the generality of our procedures and findings. We agree that group studies are important as well, but this does not mean that single case designs do not bring novelty to our study. In the new version of the manuscript, we added several information related to the previous literature that demonstrate that new investigations on BST using single case designs are warranted. Besides, single case designs involve each participant of the research as own control, considering his/her individual characteristics.
- In the new version of the manuscript, we also explained further reasons why, according to literature, new research on BST and DTT is still important.

Reviewer 3 Report
Comments and Suggestions for Authors
Overall, I think this is an interesting study in a context where the exploration of a variety of different pathways for supporting those with ASD is important. There are a few areas of tension in the literature, though, that I think need to be considered. Applied Behaviour therapy has been questioned and could be considered controversial in the literature, and with this in mind I think it necessary you pay greater attention to establishing how and why it remains a useful tool, and most importantly, when it is beneficial, and specifically who might benefit from its use. Some of the issues pertaining to AB approaches also align with broader approaches to disability, and perhaps some consideration of this would also be useful e.g, medical, model, biopsychosocial model etc. and finally, I think consideration of the view of ASD as a feature of neurodiversity also needs to be considered and discussed, and your approach situated within all of these concerns. I think it is fine to adopt the approach that you have, but the manuscript will be stronger if you acknowledge and address these tensions, and then justify why your approach is appropriate in this particular social context.
I was also left a little disappointed by your discussions section. You certainly link your findings to the extant literature in so far as describing how it corroborated this work, and at times expanded it, but I wanted more on the implications of your work. What does all this ultimately mean for practice. In your introductory section, you touched on the role of caregivers and explored why the expectations for parents etc to undertake the ABT role could be problematic, and I was hoping to see more discussion of these type of social considerations and implications.
Author Response
Dear reviewer,
Thank you very much for the important considerations and recommendations. All changes and additions in the new version of the manuscript are highlighted in red color. We also added several comments in this cover letter after your review.
- We added new references on evidence-based practices (EBPs), which also relate to several practices in the field of Applied Behavior Analysis (ABA). Scientific evidence of EBPs is systematized in literature reviews and informed through international reports. The National Professional Development Center on Autism Spectrum Disorder (NPDC) and the National Clearinghouse on Autism Evidence and Practice (NCAEP) investigate and disseminate EBP practices. The latest report, published in 2020, revealed a total of 28 EBPs, which are committed to boosting the development of individuals diagnosed with ASD through the modification of socially significant behaviors, that is, by decreasing undesired interfering behaviors and establishing more adaptive ones, including appropriate language and communication (Schmidt et al., 2024; Steinbrenner et al., 2020).
- Among the EBPs in the field of ABA, there are Behavioral Skill Training (BST) and Discrete Trial Teaching (DTT), which were of major interest in our study. Regarding these practices, we also included chapter references derived from a recent handbook of ABA-based interventions for ASD (Creem et al., 2022; Leaf, Cihon, et al., 2022; Leaf, Ferguson, et al., 2022). The literature shows that DTT benefits many learners with ASD, who demand comprehensive interventions. DTT aims at developing several skills which are commonly impaired. BST is effective in training several individuals interested in providing behavioral interventions, including DTT, to learners diagnosed with ASD. A recent meta-analysis investigation conducted by Fingerhut and Moeyaert (2022) resulted in the identification of 46 studies that used single case designs to assess the effectiveness of BST (or some of its components) to train parents, caregivers, professionals and paraprofessionals to implement DTT to children diagnosed with ASD accurately.
- It was discussed that, when the four components of BST are implemented together, they are statistically significantly effective and that, on average, the participants from the studies in the meta-analysis (Fingerhut & Moeyaert, 2022) demonstrated teaching integrity of 96.06% DTT components implemented correctly. On the other hand, it is important to mention that the studies did not include consistent information on data from the learners with ASD. In this sense, it is reasonable to assume that more research on BST is warranted especially to assess its effects on the systematic repertoire acquisition by learners with ASD. In our study, we sought to do this. We sought to fill a gap in literature.
- In the discussion section of the manuscript, we explained that DTT may benefit learners with ASD in need of comprehensive interventions, and that DTT is conducted either in a distraction-free environment for learners who lack attending skills or in a more naturalistic environment, such as a school or the residence, for learners to whom responding in a context with distractions represents an important goal. In our case, the participating children with ASD were only taught in a university laboratory room, which represents a more distraction-free environment.
- Out of the current 28 EBPs informed in the last report by NPDC and NCAEP, some are in the field of ABA. We acknowledge that practices from other areas of knowledge are important and contribute to the development of learners with ASD. This includes, for example, EBP practices on Sensory Integration and Exercise and Movement. The ABA-based practices of more interest for us, in the current study, were DTT and BST. As we said before, the literature shows scientific evidence of effectiveness for both practices, but there are still gaps that justify carrying out new studies. The meta-analysis by Fingerhut and Moeyaert (2022) indicated that more research on BST is needed to examine the impact of training on skill acquisition by the children with ASD. Plus, the meta-analysis also identified that a small percentage of the studies involved parent training. More research needs to include children with ASD with more delays, that is, children who are minimally vocal or show significant delays.
- About social validity, although this measure was consistently obtained with the university students regarding their training, that is, all of them rated the effectiveness of BST highly, social validity measures by the children’s parents or other caregivers, regarding skill development by the children, were not consistently obtained. Nevertheless, the parents and other caregivers of the participating children with ASD in this investigation informally said that the children showed improvements that reflected their functioning in their day-to-day environments. In other words, the caregivers said they were showing better attending behaviors and communicating at home and at school. We understand that this may represent an important implication of our study. It suggests that training university students in carrying out ABA interventions benefits the development of children with ASD and that this is acknowledged by their parents and other caregivers. Anyway, it is important that future studies on BST aiming to teach multiple skills also consistently take data on the social validity of the results representing skill gains in the children’s repertoire. This, in fact, represents one of the important parameters for validating research in ABA to ASD involving single-case designs (What Works Clearinghouse - WWC, 2022).
- Still in the discussion section, we discussed that more research on BST training with parents and other caregivers is also warranted, and that the meta-analysis by Fingerhut and Moeyaert (2022) indicated that few studies involved the training of parents and other caregivers. Besides, in a Country like Brazil, there are variables that make in-person training for parents and other caregivers difficult: Living in rural areas, financial difficulties in traveling to the training location and, furthermore, often not having someone to help taking care of the children during training. Also, a pandemic, such as Covid-19, makes in-person training difficult. An alternative to the limitations described so far may be the use of telehealth to train parents and other caregivers. We refined the discussions about some recent studies that investigated the effectiveness of a remote BST (through telehealth) with internet connections to train parents to implement DTT to their children with ASD (Azzano et al., 2022; Ávila & Matos, 2023; Higgins et al., 2023). In these studies, the parents as participants were trained through videoconferences using computers or laptops. Remote BST successfully trained all parents to implement DTT to their children with ASD accurately. However, we also discussed that in a country such as Brazil, access to computers and laptops is difficult for many people, especially in the Northeast region for financial reasons. So, we discussed that a possible alternative to this may be remote BST training using devices such as smartphones, which most of the population owns. We argued that more research is warranted on remote BST training involving smartphones, which tend to be more accessible. Future research in this direction may involve several participants as learners, such as parents, other caregivers, professionals and paraprofessionals. These investigations may verify, for example, if remote BST through the use smartphones successfully trains several individuals in the provision of DTT to teach multiple skills to learners with ASD.
- We also added discussions on recent systematic review study by De Souza et al. (2025), which identified 59 research articles published in Brazilian and international journal from 2007 to 2024, and that investigated the effectiveness of single case methodology on the modification of socially important behaviors of individuals with ASD. De Souza et al. (2025) also assessed the quality of the studies based on the indicators of the “What Works Clearinghouse” (WWC, 2022). It was noticed that, out of 59 articles, only 14 fully followed the WWC standards.
- Our study was conducted in Brazil. We followed the standards for single-subject studies by WWC (2022), seeking to address a problem indicated in the systematic review by De Souza et al. (2025), in the sense that many Brazilian studies from the review did not follow the WWC standards. In other words, in the current investigation, data were presented in graph and table format; the independent variable (IV – BST) was systematically manipulated; interobserver agreement (IOA) was obtained by two observers across all experimental conditions and was determined above 20% of the sessions per condition; the IOA level between the observers was determined above 80% across the experimental conditions; it was used a three-tier multiple baseline for each of the two triads of university students who participated; each of the study’s phases involved at least three data points with little or no variability; also, the research involved the assessment of generalization, maintenance and social validity of procedures and results. Everything was done with the aim of ensuring the internal and external validity of procedures and results, which is expected from single case studies.
- Regarding all evidence discussed so far, we do believe that ABA procedures are beneficial for socially relevant behavior change of individuals diagnosed with ASD. However, recent meta-analysis (Fingerhut & Moeyaert, 2022) and systematic review (De Souza et al., 2025) show that more investigations are important (specially in a country as Brazil where research involving single case designs and ASD has shown an increasing trend in the last 10 years). Further investigations are warranted on more prevalent behaviors in individuals with ASD, such as interfering behaviors, communication skills at a more basic level and functional behaviors related to independence in the community. There is also a lack of research aimed at designing behavioral interventions for adolescents and adults. Research needs to include more social validity methods to precisely measure the acceptance level of interventions and results.
- We understand that there are several possibilities for further research, but the case that caught our attention the most refers to the need for further investigations into BST involving learners with ASD who are minimally vocal or who show significant delays in repertoire (Creem et al., 2022; Fingerhut & Moeyaert, 2022; De Souza et al., 2025). Research on ABA has focused on more skilled learners with ASD.
